# Endocannabinoid signaling enhances visual responses through modulation of intracellular chloride levels in retinal ganglion cells

Loïs S Miraucourt[1], Jennifer Tsui[1,2], Delphine Gobert[1], Jean-François Desjardins[3], Anne Schohl[1], Mari Sild[1], Perry Spratt[1,4], Annie Castonguay[5], Yves De Koninck[5], Nicholas Marsh-Armstrong[6,7], Paul W Wiseman[3], Edward S Ruthazer[1]*

[1]Montreal Neurological Institute, McGill University, Montreal, Canada; [2]Department of Biology, University of La Verne, La Verne, United States; [3]Department of Physics, McGill University, Montreal, Canada; [4]Neuroscience Graduate Program, University of California, San Francisco, San Francisco, United States; [5]Institut Universitaire en santé mentale de Québec, Université Laval, Québec, Canada; [6]Department of Neuroscience, Johns Hopkins University School of Medicine, Baltimore, United States; [7]Kennedy Krieger Institute, Baltimore, United States

*For correspondence: edward.ruthazer@mcgill.ca

**Abstract** Type 1 cannabinoid receptors (CB1Rs) are widely expressed in the vertebrate retina, but the role of endocannabinoids in vision is not fully understood. Here, we identified a novel mechanism underlying a CB1R-mediated increase in retinal ganglion cell (RGC) intrinsic excitability acting through AMPK-dependent inhibition of NKCC1 activity. Clomeleon imaging and patch clamp recordings revealed that inhibition of NKCC1 downstream of CB1R activation reduces intracellular $Cl^-$ levels in RGCs, hyperpolarizing the resting membrane potential. We confirmed that such hyperpolarization enhances RGC action potential firing in response to subsequent depolarization, consistent with the increased intrinsic excitability of RGCs observed with CB1R activation. Using a dot avoidance assay in freely swimming *Xenopus* tadpoles, we demonstrate that CB1R activation markedly improves visual contrast sensitivity under low-light conditions. These results highlight a role for endocannabinoids in vision and present a novel mechanism for cannabinoid modulation of neuronal activity through $Cl^-$ regulation.

## Introduction

Endocannabinoids (eCBs) are lipid-derived regulators of synaptic transmission that serve as important mediators of functional plasticity (*Bellocchio et al., 2013*; *Bender et al., 2006*; *Clapper et al., 2010*; *El Manira and Kyriakatos, 2010*; *Han et al., 2012*; *Kato et al., 2012*; *Sjöström et al., 2003*) and nervous system development (*Argaw et al., 2011*; *Berghuis et al., 2007*; *Harkany et al., 2008*; *Li et al., 2009*). In the retina, high levels of type 1 cannabinoid receptors (CB1R), as well as the eCB production and degradation machinery, have been found in all vertebrate models tested, and within nearly all retinal cell types throughout development and adulthood (*Straiker et al., 1999*; *Yazulla, 2008*; *Zabouri et al., 2011*). Although eCB modulation of voltage-dependent $Ca^{2+}$ and $K^+$ conductances, typically resulting in the downregulation of neurotransmitter release, has been demonstrated at multiple stages of retinal processing from photoreceptors to retinal ganglion cells (RGCs) (*Fan and Yazulla, 2003*; *Lalonde et al., 2006*; *Middleton and Protti, 2011*; *Yazulla et al., 2000*), these observations have not been directly linked to specific perceptual outcomes. On the

other hand, cannabis consumption in humans has been reported to positively modulate contrast discrimination (*Russo et al., 2004*; *West, 1991*). Given the overall importance of eCB signaling in modulating neuronal activity, we set out to test whether retinal eCB signaling impacts visual function, and if so, by what modes of action.

The transparency and amenability to whole-cell recording of the *Xenopus laevis* tadpole permits the application of both functional imaging and electrophysiological recording in the visual system of the intact animal. Exploiting the strengths of this model, we demonstrate a novel mechanism by which eCB signaling increases the intrinsic excitability of RGCs through AMPK-dependent inhibition of the $Na^+$-$K^+$-$2Cl^-$ co-transporter 1 (NKCC1). NKCC1 inhibition causes a reduction in intracellular $Cl^-$ levels, which enhances tonic glycinergic currents and hyperpolarizes the resting membrane potential in RGCs. We show that membrane hyperpolarization actually makes RGCs fire more spikes in response to subsequent stimulation. Accordingly, we found that activation of CB1Rs produces a marked reduction of the contrast threshold needed to trigger a visually evoked escape behavior by tadpoles under dim light conditions, consistent with enhanced visual perception.

## Results

### CB1R is present in the retina of *Xenopus laevis* tadpoles

The presence of CB1Rs has been reported in the brain of the adult *Xenopus laevis* frog (*Cesa et al., 2001*; *Cottone et al., 2003*) and in the tadpole olfactory bulb (*Breunig et al., 2010*; *Czesnik et al., 2007*). We examined the retinae of *Xenopus* tadpoles, and found intense CB1R immunoreactivity (CB1R-IR) in the outer and inner plexiform layers (*Figure 1a,b*; *Figure 1—figure supplement 1*), consistent with reports in other species (*Middleton and Protti, 2011*; *Yazulla et al., 2000*; *Zabouri et al., 2011*). Somatic CB1R-IR has been described in the RGCs of salamander and chick (*Straiker et al., 1999*), but not in rat (*Yazulla et al., 1999*). Using Isl2b:GFP transgenic frogs, in which expression of green fluorescent protein (GFP) in the retina is restricted to RGCs, we found CB1R-IR associated with GFP-positive RGCs in histological sections (*Figure 1c*) and in overnight dissociated retinal cultures (*Figure 1d*). These data show that CB1R-IR is present at multiple levels, including in RGCs within the retina of the tadpole, consistent with a role for the eCB system in early visual processing.

### CB1R activation increases RGC firing in response to visual stimulation

Extracellular multi-unit recordings in isolated eye preparations (*Figure 2a*) revealed that application of the CB1R agonist WIN 55,212-2 (1 µM) increased spiking rates of RGCs in response to both full field light-ON (before: 34.2 ± 3.1 Hz, after: 43.3 ± 4.6 Hz, n = 10, p=0.010, two-way RM ANOVA) and light-OFF (before: 37.2 ± 3.1 Hz, after: 45.2 ± 4.2 Hz, n = 10, p=0.011) stimuli (*Figure 2b,c*; *Figure 2—figure supplement 1a,d*). This cannabinoid-mediated enhancement of evoked RGC responses was confirmed using a different CB1R agonist arachidonyl-2'-chloroethylamide (ACEA, 1 µM) which gave a similar enhancement of spiking to light-OFF stimuli (*Figure 2d*; *Figure 2—figure supplement 1b,e*; before: 30.5 ± 4.1 Hz, after: 40.0 ± 6.0 Hz, n = 10, p=0.033). To assess whether endogenous cannabinoids produced by the retina can generate this effect in the absence of exogenous ligand, we applied URB597 (2 µM), a selective inhibitor of fatty acid amide hydrolase (FAAH), the enzyme that degrades the eCB anandamide (AEA). URB597 treatment caused elevated spiking rates in response to light-ON (before: 22.8 ± 5.3 Hz, after: 31.3 ± 6.6 Hz, n = 10, p=0.049) and light-OFF stimulation (*Figure 2e*; *Figure 2—figure supplement 1c,f*; before: 51.0 ± 7.0 Hz; after: 61.7 ± 10.3 Hz, n = 10, p=0.031), suggesting that AEA synthesized in the retina is able to modulate visual responses.

CB1R activation could act by modulating visual responses in retinal cells upstream of RGCs or by directly increasing RGC excitability. We first performed electroretinogram (ERG) recordings in the isolated eye preparation to test if WIN 55,212-2 application modulated responses in the outer layers of the retina (*Figure 2f*). ERGs revealed visual responses of various non-spiking cell types in the outer retina, including photoreceptors (a-wave), ON-center bipolar cells (b-wave) and OFF-center bipolar cells (d-wave). Visual stimulation produced stable ERG responses that were generally more robust for light-OFF stimuli, consistent with previous reports in the *Xenopus* visual system (*Engert et al., 2002*; *Zhang et al., 2000*). However, perfusion of WIN 55,212-2 did not produce any

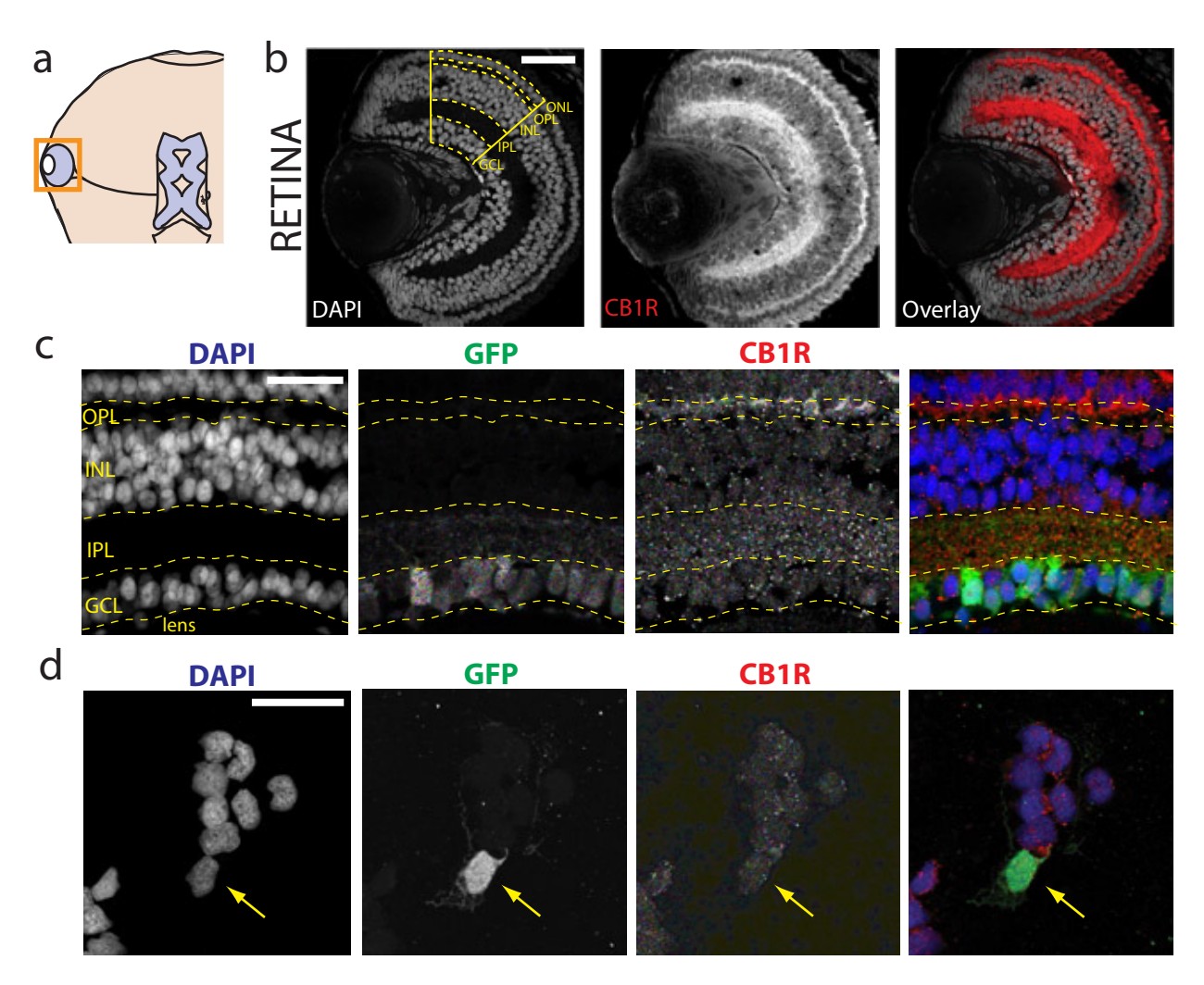

**Figure 1.** Immunolocalization of CB1R in the Xenopus laevis tadpole eye. (**a**) Cartoon of tadpole retinotectal system. (**b**) DAPI (gray) and CB1R-IR (red) co-labeling of a retinal cryosection. Dashed lines highlight ganglion cell layer (GCL), inner plexiform layer (IPL), inner nuclear layer (INL), outer plexiform layer (OPL) and outer nuclear layer (ONL) regions. (**c**) DAPI (blue), GFP-expressing retinal ganglion cells (green, Isl2b:EGFP transgenic) and CB1R-IR (red) in histological section of the retina. (**d**) Cell culture of dissociated cells from retina of isl2b:GFP animals. DAPI (blue), GFP (green) and CB1R-IR (red). Scale bar = 100 μm in **b** and 25 μm in **c**,**d**.

The following figure supplement is available for figure 1:

**Figure supplement 1.** Cells of the *Xenopus* retina.

significant changes in the amplitudes of the a-, b- or d-waves in response to step increments or decrements of light (*Figure 2g*, n = 6, 6, 6, 6; p=0.78, 0.79, 0.89, 0.68, two-way RM ANOVA), suggesting that CB1R activation does not impact light-evoked activity in the outer part of the retina, consistent with a recent report in the mouse (*Cécyre et al., 2013*).

Local activation of CB1Rs at bipolar axon terminals could potentially modify $Ca^{2+}$ influx to modulate the release of glutamate onto RGC dendrites. Similarly, CB1Rs at inhibitory amacrine presynaptic contacts could alter $Ca^{2+}$ influx to modulate the release of GABA or glycine. Such effects would go undetected in ERG recordings but would alter the spiking output of RGCs. To test these possibilities, we used retinal electroporation to express the genetically-encoded $Ca^{2+}$ reporter GCaMP6s in bipolar and amacrine cells to measure $Ca^{2+}$ levels in vivo at their presynaptic terminals in response to light flashes (*Figure 3a–i*). Light stimulation reliably evoked strong $Ca^{2+}$ transients in individual

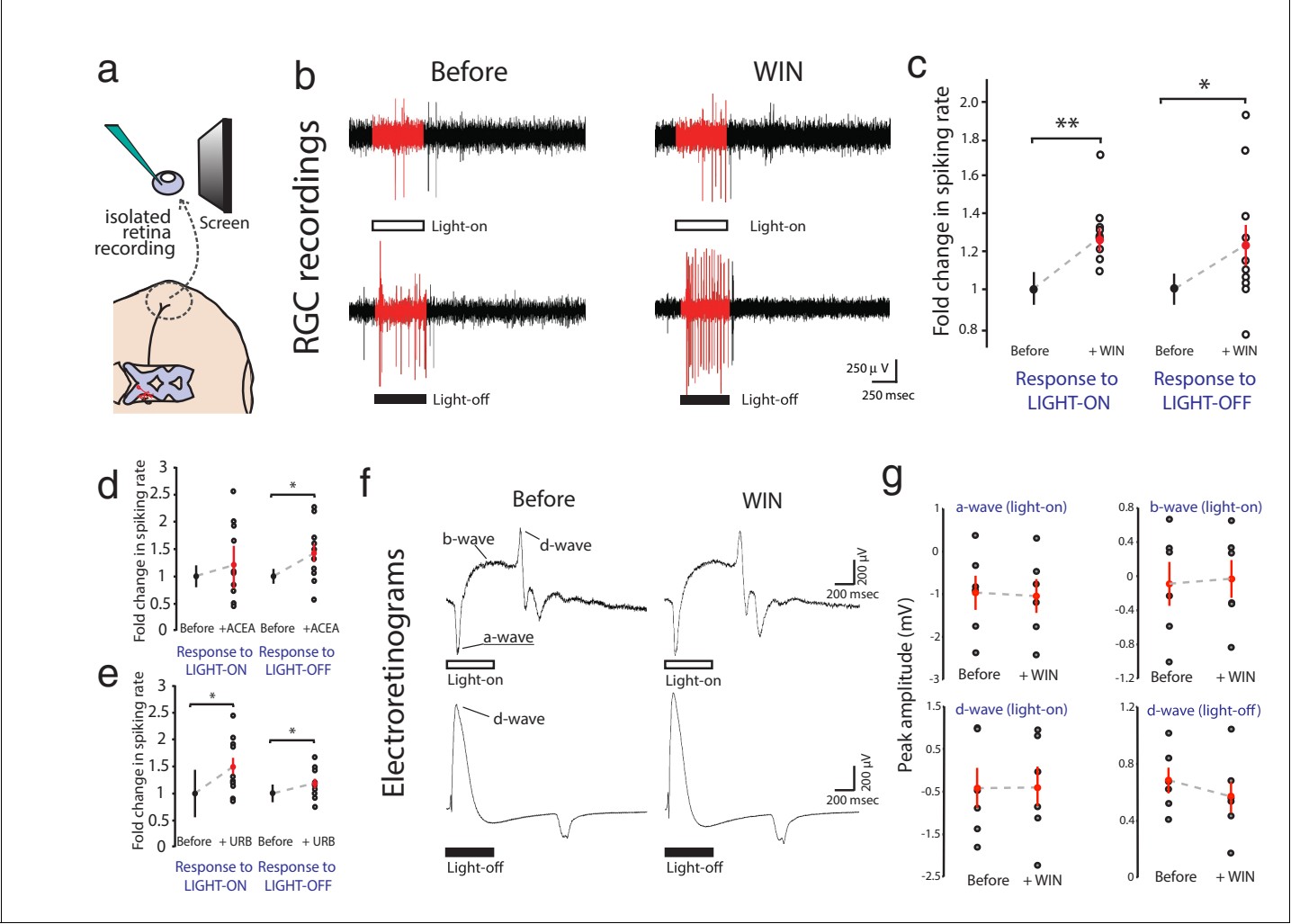

**Figure 2.** WIN 55,212-2 enhances visually evoked firing of RGCs without impacting activity in the outer retina. (**a**) Isolated eye preparation for extracellular retinal recordings. (**b**) Representative extracellular multi-unit RGC spike recordings in the isolated eye preparation in response (red) to 500 ms full-field light-ON (top) and light-OFF (bottom) stimuli, before and after application of the CB1R agonist WIN 55,212-2. (**c**) WIN 55,212-2 application increases RGC spiking rates to stimuli (n = 10 animals). (**d**) A different CB1R agonist ACEA also increases evoked firing to light-OFF stimuli (n = 10 animals, p=0.42 for light-ON). (**e**) Elevating endogenous AEA by application of URB597 also enhances RGC responiveness (n = 10 animals). (**f**) ERGs were used to measure outer retina responses to 500 ms light-ON (top) and light-OFF (bottom) flashes before (control) and after WIN 55,212-2 application. No change was observed. (**g**) Peak amplitudes for each component of the ERG in response to light-ON (n = 6 animals) and light-OFF (n = 6 animals) stimulation were not significantly affected by application of WIN 55,212-2. *p<0.05, **p<0.01, two-way RM ANOVA with Holm-Sidak posttest. ACEA, Arachidonyl-2'-chloroethylamide; AEA, anandamide; ERG, Electroretinogram; RGC, Retinal ganglion cell.

The following figure supplement is available for figure 2:

**Figure supplement 1.** Individual experiments for *Figure 2c–e*, and ERG components.

bipolar axon terminals (*Figure 3b*). Comparing average peak Ca$^{2+}$ transients evoked by full-field light-ON and light-OFF stimulation before and after WIN 55,212-2 perfusion (*Figure 3c*), we found that CB1R activation did not increase bipolar axon Ca$^{2+}$ transient amplitude in response to visual stimulation, but instead decreased it in the case of light-OFF stimulation (*Figure 3d,e*, ON: n = 10, p=0.17, OFF: n = 10, p=0.002), consistent with reports from salamander of a CB1R-mediated reduction of the L-type Ca$^{2+}$ current[12]. Thus, its effects on Ca$^{2+}$ at bipolar cell terminals appear more likely to reduce than to increase RGC spiking. Peak Ca$^{2+}$ transients in amacrine cell terminals evoked by

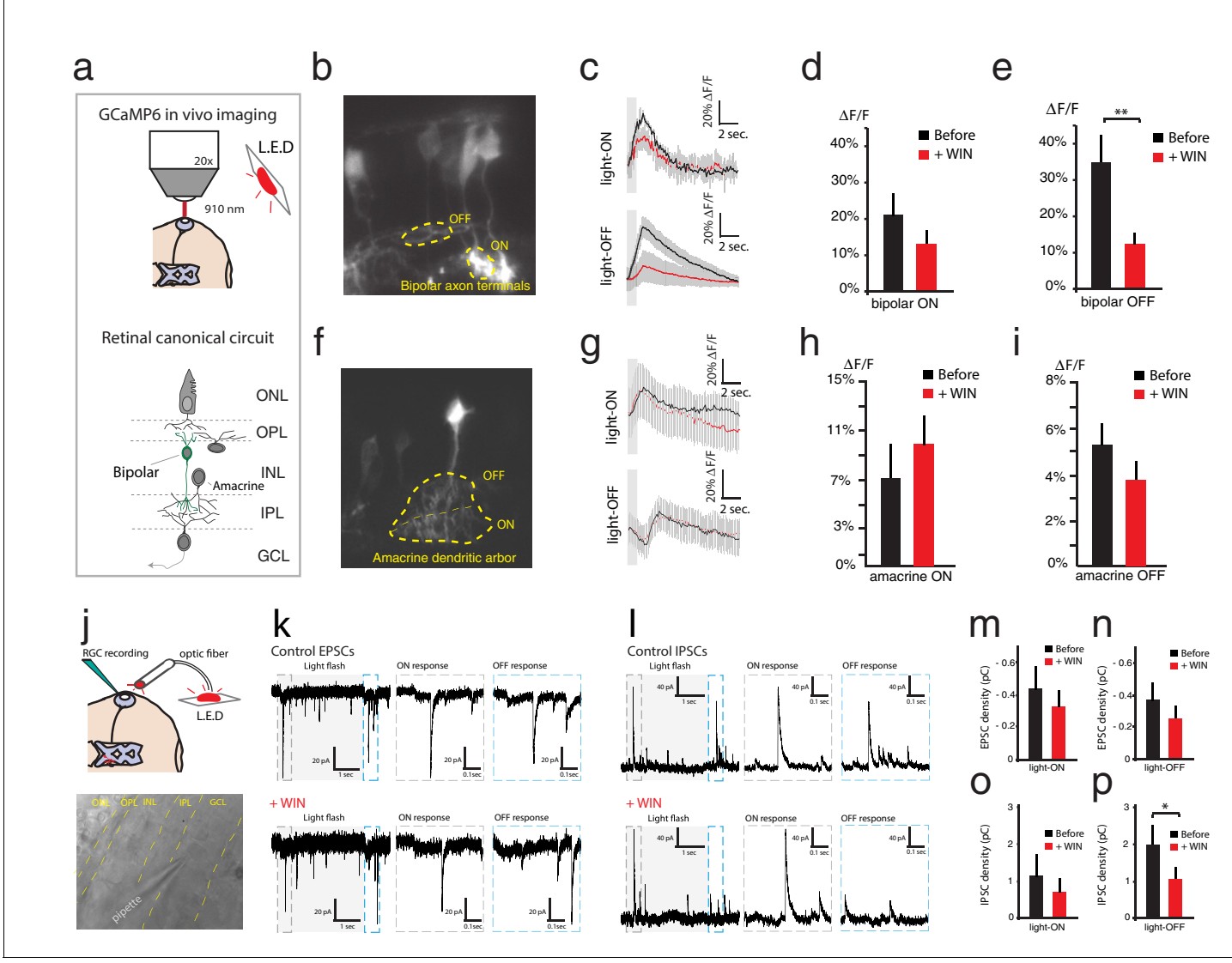

**Figure 3.** WIN 55,212-2 reduces some synaptic input strength onto RGCs. (**a–i**) In vivo imaging of GCaMP6s expressed in the retina was used to measure Ca²⁺ influx at synaptic terminals of bipolar and amacrine cells. (**a**) Schematic representation of the imaging configuration and retinal circuitry showing bipolar and amacrine cell type locations. (**b**) GCaMP6s expressed in bipolar cells, with ROIs exemplifying ON- and OFF-specific bipolar axon terminals. (**c**) Average calcium responses of individual bipolar axon terminal to 10 light-ON (top) or 10 light-OFF (bottom) flashes before (black) and after (red) WIN 55,212-2 perfusion. (**d**) △F/F peak amplitudes in bipolar axon terminals (n = 10 terminals from four animals) to light-ON before and after WIN 55,212-2 perfusion (p=0.17). (**e**) A significant change in △F/F peak amplitude of bipolar axon terminals (n = 10 terminals from four animals) was observed to light-OFF following WIN 55,212-2 perfusion. (**f**) GCaMP6s expressed in amacrine cells, with ROI exemplifying ON- and OFF-specific terminals. (**g**) Average calcium responses of amacrine cell terminals to 10 light-ON (top) or 10 light-OFF (bottom) flashes before (black) and after (red) WIN 55,212-2 infusion. (**h**) No change (p=0.36) in △F/F peak amplitude was observed in amacrine cell terminals (n = 10) to light-ON before and after WIN 55,212-2 infusion. (**i**) △F/F peak amplitude in amacrine cell terminals (n = 10) to light-OFF stimulation was also unchanged (p=0.48) by WIN 55,212-2 perfusion. (**j–p**) Voltage-clamp recordings of RGCs in vivo (n = 7) during light stimulation. (**j**) Schematic representation of whole cell RGCs recording configuration with light stimulation driven by red LED flashes conveyed to the eye through an optic fiber. (**k**) left: Voltage clamp raw trace recordings of EPSCs from an RGC held at −65 mV, evoked in response to a 3 s light flash. Right: inset of the raw trace showing fast inward currents in response to light. (**l**) left: Trace of IPSCs from RGC held at 0 mV, evoked in response to a 3 s light flash. Right: inset of the raw trace showing fast outward currents in response to light. (**m–n**) Average total integrated inward current responses to light-ON (**m**) or light-OFF (**n**) flashes before (black) and after (red) WIN 55,212-2 perfusion with RGCs held at −65 mV. N = 10, p=0.025 for simple effect of WIN application by two-way ANOVA. (**o–p**) Average total integrated outward current responses to light-ON (**o**) or light-OFF (**p**) flashes before (black) and after (red) WIN 55,212-2 addition with RGCs held at 0 mV. n = 10, *p<0.05, **p<0.01, two-way RM ANOVA with Holm-Sidak post-test. EPSC, Excitatory postsynaptic current; IPSC, Inhibitory postsynaptic current; RGC, Retinal ganglion cell; ROI, Regions of interest.

light-ON and light-OFF stimulation were not significantly different before and after WIN 55,212-2 perfusion (*Figure 3f–i*, ON: n = 10, p=0.36; OFF: n = 10, p=0.48).

Effects on synaptic inputs to RGCs were also analyzed by in vivo whole-cell voltage clamp recordings from RGCs, held at −65 mV and 0 mV to reveal excitatory (EPSC) and inhibitory postsynaptic currents (IPSC) evoked by visual stimulation, respectively (*Figure 3j–p*). Application of WIN 55,212-2 resulted in a significant decrease in the total integrated EPSC, measured over 1 s following stimulus onset (*Figure 3m,n*, n = 7, p=0.025, two-way RM ANOVA, before vs. +WIN simple effect). Inhibitory synaptic input in response to light-ON stimuli was unchanged by WIN 55,212-2 (*Figure 3o*, n = 5, p=0.199, two-way RM ANOVA with Holm-Sidak posthoc test) but a decrease in the total integrated IPSC in response to light-OFF stimulation was observed (*Figure 3p*, n = 5, p=0.029). Taken together these data indicate that, cannabinoid modulation of excitatory synaptic inputs onto RGCs is, if anything, acting in the wrong direction to enhance firing in response to visual stimuli. On the other hand it is possible that disinhibition may in part contribute to this phenomenon.

An alternative mechanism by which cannabinoids might enhance visually evoked firing rates would be to directly elevate the intrinsic excitability of RGCs, making it easier for them to fire action potentials. The simplest way this might occur would be through depolarization of the resting membrane potential closer to the threshold for firing. However, studies in salamander retina have suggested that membrane depolarization can cause sodium channel inactivation that paradoxically hinders action potential generation (*Weick and Demb, 2011*). Thus, the cannabinoid-mediated enhancement of RGC responsiveness could potentially be due one or a combination of synaptic disinhibiton, mechanisms that depolarize or that hyperpolarize resting membrane potential.

## CB1R-mediated enhancement of RGC firing requires glycinergic transmission and NKCC1 activity

To elucidate these mechanisms, we examined the effects of various pharmacological manipulations on the cannabinoid-mediated modulation of responses to full-field light-OFF (*Figure 4*) or light-ON stimuli (*Figure 4—figure supplement 1*). Pretreatment with the CB1R blocker AM-251 (5 μM) completely prevented the potentiating effects of WIN 55,212-2 (*Figure 4a,* n = 10, p=0.487, one-tailed one-way RM ANOVA) or ACEA (*Figure 4b*, n = 10, p=0.360) on RGC firing, further underscoring the requirement for CB1R activation. The enhancement produced by elevating endogenous AEA using URB597 (*Figure 2e*) was also prevented by AM-251 (*Figure 4c*, n = 10, p=0.500). The selective GABA-A receptor blocker gabazine (6 μM) had no specific effect on visually-evoked spike rates and did not prevent the WIN 55,212-2-induced increase in RGC firing in response to light (*Figure 4d*, n = 10, p=0.012). However, strychnine (60 μM), a potent and selective antagonist of glycine receptors (GlyRs) prevented (*Figure 4e*, n = 10, p=0.423) and reversed (*Figure 4f*, n = 10, p=0.044) the enhancement caused by WIN 55,212-2 application. These results show that both CB1Rs and GlyRs are required for the cannabinoid-mediated increase in RGC response to visual stimulation. We further investigated whether this synergy could be due to a CB1R-dependent regulation of the gradient of [Cl$^-$] across the plasma membrane, which provides the driving force for glycinergic current.

Intracellular Cl$^-$ is regulated in most neurons by the coordinated activities of NKCC1 and the K$^+$-Cl$^-$ cotransporter 2 (KCC2) which shuttle Cl$^-$ ions into and out of the cell respectively (*Ben-Ari, 2002*). Previous studies have identified the regulation of [Cl$^-$]$_i$ levels by NKCC1 activity as a mechanism mediating inhibitory plasticity in immature hippocampal neurons (*Balena and Woodin, 2008*). NKCC1 and KCC2 are co-expressed in RGCs in fish retina (*Dmitriev et al., 2007*), and we were able to detect both mRNAs in stage 45 *Xenopus* retina (*Figure 4i*). Perfusion of the selective NKCC1 inhibitor bumetanide (10 μM) alone was sufficient to increase the firing rates of RGCs in response to light, mimicking the effect of CB1R activation (*Figure 4g*, n = 10, p=0.025). Subsequent addition of WIN 55,212-2 in the presence of bumetanide produced no additional increase in RGC firing rate (n = 10, p=0.224). Furthermore, WIN 55,212-2 application (*Figure 4h*, n = 9, p=0.012) occluded the increase in RGC firing rate produced by adding bumetanide (*Figure 4h*, n = 9, p=0.224 vs. WIN alone). These results argue that CB1R activation may work by regulating NKCC1 function to modulate the driving force of chloride currents in RGCs.

NKCC1 activity can be regulated through serine/threonine phosphorylation of its intracellular domains. In mammals, the WNK-activated Sterile-20 (Ste20)-related Proline/Alanine-rich Kinase (SPAK) and Oxidative Stress-Response Kinase 1 (OSR1) have been shown to stimulate human NKCC1 through phosphorylation at T203, T207, T212 and T217 (*Darman et al., 2001*;

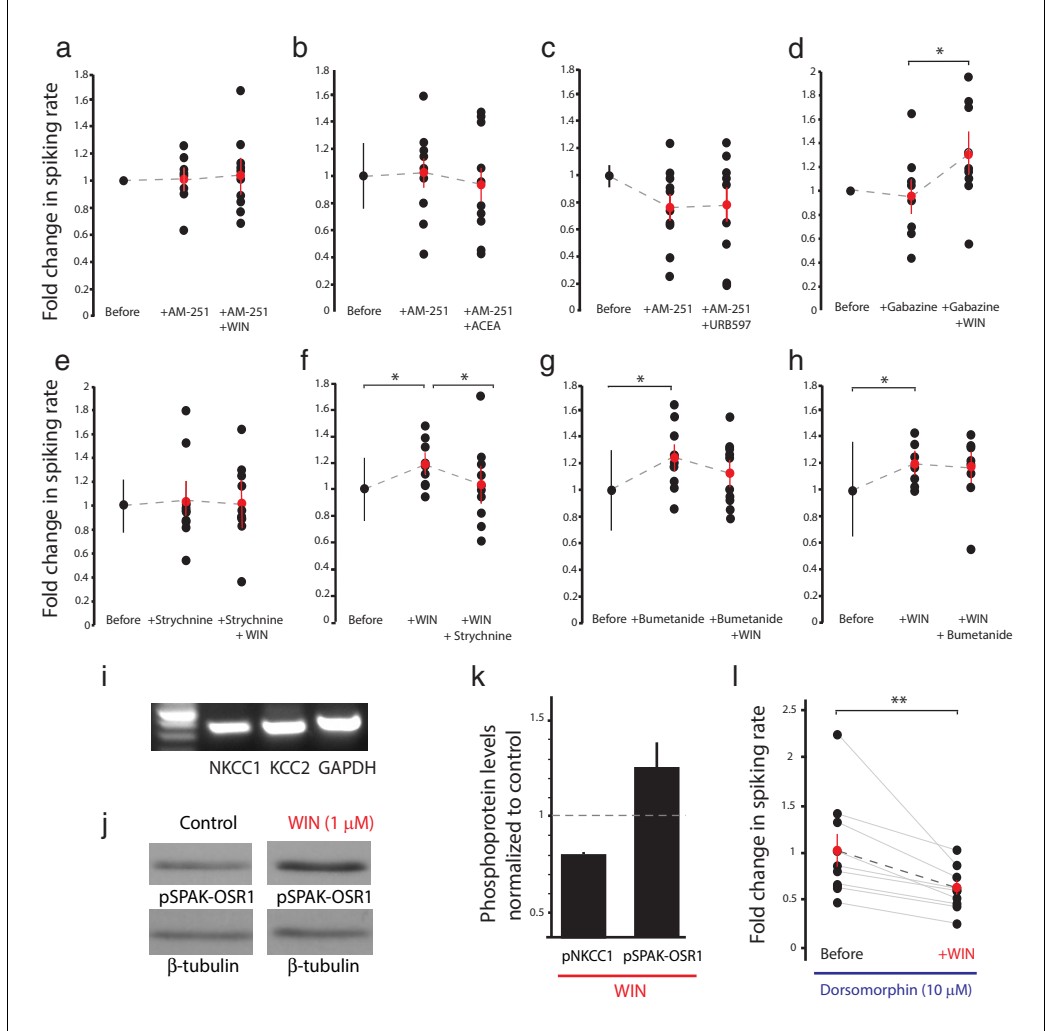

**Figure 4.** CB1R-mediated increase in RGC responsiveness requires GlyRs and NKCC1 inhibition by AMPK. (a–h) Extracellular multi-unit spike rates in RGCs were evoked with a 500 ms light-OFF stimulus. After establishing baseline response rates (Before), various pharmacological agents were washed on sequentially. (a–c) The CB1R inverse agonist AM-251 prevented (a) WIN 55,212-2-, (b) ACEA- and (c) URB597-mediated increases in spiking rates (n = 10 animals for each group), confirming the requirement for CB1R activation. (d) The GABA-A receptor blocker gabazine did not prevent the enhancement of RGC excitability, but the selective GlyR antagonist strychnine completely abolished the enhancement when applied either (e) before or (f) after WIN 55,212-2. (g) The NKCC1 blocker bumetanide mimicked and occluded the effects of WIN 55,212-2, and (h) no additional effect was observed if bumetanide was applied after WIN 55,212-2 (n = 10 animals for **d–g**, n = 9 animals for **h**). (i) RT-PCR confirmed expression of the cation-chloride cotransporters NKCC1 and KCC2 in the *Xenopus* retina at stage 45. (j) Western blots show a 60 kD band staining for phospho-SPAK/phospho-OSR1, and β-tubulin, in controls or following treatment with WIN-55,212-2 (1 µM) in the *Xenopus* retina at stage 45. (k) Bar graph showing fold-change in phosphoprotein levels following treatment with WIN-55,212-2 (1 µM) of $NKCC1_{phospho\ Thr212+Thr217}$, and phospho-SPAK/phospho-OSR1 normalized to control levels (n = 4). (l) In the presence of dorsomorphin (10 µM) to block AMPK, WIN-55,212-2 (1 µM) treatment no longer enhances, but instead reduces RGC firing rates (n = 10 animals). (a–h) *p<0.05 one-tailed RM ANOVA with Holm-Sidak posttest, (l) **p<0.01, paired t-test. RGC, Retinal ganglion cell.

The following figure supplement is available for figure 4:

**Figure supplement 1.** Similar trend of pharmacological agents on light-ON responses to that observed for light-OFF stimuli.

---

*Richardson et al., 2008*; *Vitari et al., 2006*), whereas AMP-dependent kinase (AMPK) inhibits NKCC1 by phosphorylation at S77 (*Fraser et al., 2014*). We examined the involvement of these regulatory molecules by performing western blotting on retina homogenates from animals treated with WIN 55,212-2. WIN 55,212-2 application produced a striking increase in the levels of phosphorylated SPAK/OSR1, indicating its activation downstream of the CB1R (*Figure 4j,k*); however, this was not

accompanied by an increase in phospho-NKCC1$^{T212/T217}$ (*Figure 4—figure supplement 1i*). Thus, SPAK/OSR1, although activated, appears not to phosphorylate and activate NKCC1 under these conditions. Moreover, our physiology data would have predicted *decreased* NKCC1 activity. We therefore wondered whether AMPK activation might be suppressing NKCC1 function. Because no phospho-specific antibody against the AMPK site S77 on NKCC1 is available, we instead tested this hypothesis functionally by using the selective AMPK inhibitor dorsomorphin. We found that dorsomorphin (10 µM) reversed the effects of WIN 55,212-2, reducing RGCs spiking rate in response to light-OFF flashes (before: 37.5 ± 6.3 Hz, after: 22.9 ± 2.6 Hz, *Figure 4l*, n = 10, p=0.0095, paired t-test). These results indicate that CB1R signaling inhibits NKCC1 through the activation of AMPK.

## CB1R activation lowers [Cl$^-$]$_i$ specifically in RGCs

The inhibition of NKCC1 in RGCs should manifest as a reduction in [Cl$^-$]$_i$. To measure changes in [Cl$^-$]$_i$ we performed in vivo imaging of retinae electroporated to express a modified Cerulean(CFP)-Topaz(YFP) version of Cl$^-$ reporter Clomeleon (*Kuner and Augustine, 2000*), and made ratiometric measurements of Föster resonance energy transfer (FRET) (*Figure 5a*). Clomeleon-expressing RGCs, bipolar cells, amacrine cells and Müller glia were identified based on morphology and location. RGCs could be distinguished from displaced amacrine cells in the ganglion cell layer by the presence of axonal processes. Baseline two-photon z-stacks of cyan and yellow fluorescence were collected every 2 min for 10 min, before and after the onset of WIN 55,212-2 perfusion at 10 min, as in the example shown in *Figure 5b* and quantified in *Figure 5c*. In all experiments, CB1R activation consistently caused a rapid decrease in the cyan:yellow emission ratio in RGCs (*Figure 5d,e*, n = 15 cells, p=0.001, two-way RM ANOVA with Holm-Sidak post-test) indicating a reduction in [Cl$^-$]$_i$. No FRET changes in amacrine (n = 16, p=0.649), bipolar (n = 12, p=0.972) or Müller glia (n = 8, p=0.997) cell types were detected (*Figure 5e*). The decrease in the cyan:yellow emission ratio in RGCs was not observed when expressing a control Cl$^-$-insensitive Clomeleon-M69Q construct (*Figure 5f*, n = 5, p=0.996). Pretreatment with AM-251 (n = 8) or bumetanide (n = 9) prevented any changes in [Cl$^-$]$_i$ upon addition of WIN 55,212-2 (*Figure 5f*, p=0.996 for both). These data suggest that CB1R activation induces a RGC-specific decrease in [Cl$^-$]$_i$ by regulating NKCC1 activity. To determine if synaptic activity might contribute to the changes in [Cl$^-$]$_i$, we blocked synaptic transmission with a cocktail of DNQX (20 µM), AP4 (50 µM), picrotoxin (100 µM), strychnine (60 µM) and TTX (1 µM). With synaptic block, WIN 55,212-2 application still significantly decreased the cyan:yellow ratio in RGCs (*Figure 5g*, n = 14, p<0.001). These data indicate that retinal CB1R activation rapidly decreases [Cl$^-$]$_i$ specifically in RGCs through a NKCC1-dependent and neurotransmission-independent mechanism.

## CB1R-mediated Cl$^-$ extrusion hyperpolarizes RGCs and increases intrinsic excitability

We next sought to electrophysiologically confirm the cannabinoid-mediated decrease in [Cl$^-$]$_i$ by testing whether WIN 55,212-2 application induced a negative shift in the reversal potential of glycinergic currents (E$_{Gly}$) in RGCs in vivo (*Figure 6a*). Ideally, gramicidin-perforated patch clamp recordings would be used to prevent Cl$^-$ in the recording pipette from disrupting intracellular concentrations; however, this technique proved to be too difficult for in vivo RGC recordings. We therefore followed the example of *Ferrini et al. (2013)* who used conventional whole-cell voltage clamp recordings to detect Cl$^-$ extrusion from spinal neurons (*Ferrini et al., 2013*). RGCs were filled through the recording pipette with a fluorescent dye to confirm their identity based on their dendritic morphologies and the presence of an axonal projection to the optic tectum (*Figure 6b–d*). We measured E$_{Gly}$ by generating current-voltage (I-V) plots of peak current evoked by a brief application of glycine (500 µM for 20 ms) over a range of holding potentials (*Figure 6e*). A small but significant leftward shift for E$_{Gly}$ could be seen following wash-on of WIN 55,212-2 in the bath (ΔV = −3.2 ± 0.8 mV, n = 7, p=0.008). Inhibition of NKCC1 with bath application of bumetanide produced a similar shift in E$_{Gly}$ (*Figure 6f*, ΔV = −1.6 ± 0.5 mV, n = 5, p=0.026). Thus, electrophysiological recordings confirm that WIN 55,212-2 lowers intracellular Cl$^-$.

Based on these findings, we predicted that Cl$^-$ currents in RGCs, such as those mediated by GlyRs, should hyperpolarize the RGC membrane potential in response to CB1R activation. We tested this prediction by performing gramicidin-perforated patch recordings on dissociated cultured RGCs.

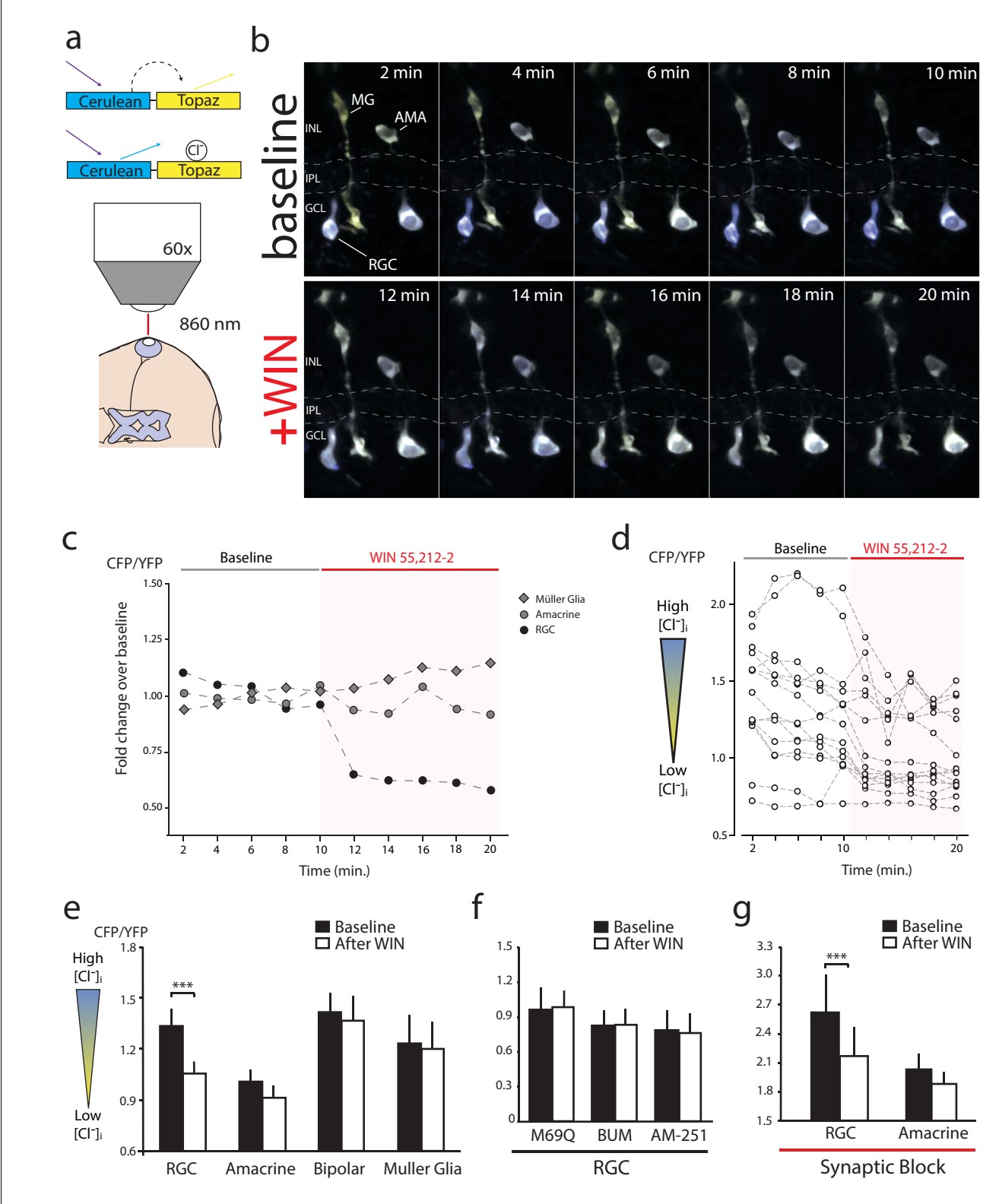

**Figure 5.** WIN 55,212-2 decreases intracellular Cl⁻ in RGCs. (a) Cerulean(CFP)-Topaz(YFP) modified version of the FRET-based Cl⁻ indicator clomeleon provides a ratiometric readout of changes in intracellular Cl⁻ concentration due to the quenching of YFP fluorescence by Cl⁻. (b) Example time series of in vivo clomeleon-expressing retinal cells before (top) and after (bottom) WIN 55,212-2 application. Only cells with a stable baseline CFP/YFP ratio (<10% drift) were analyzed. (c) Somatic measurements of CFP/YFP from the cells in panel b, reveal an RGC-specific change in the CFP/YFP fluorescence

*Figure 5 continued on next page*

Miraucourt *et al.* eLife 2016;5:e15932. DOI: 10.7554/eLife.15932

*Figure 5 continued*

ratio in response to WIN 55,212-2. (**d**) Measurements of WIN 55,212-2-induced effects on CFP/YFP fluorescence ratios in all 15 RGCs imaged in seven tadpoles. (**e**) Mean CFP/YFP ratios in RGCs (n = 15), amacrine cells (n = 16), bipolar cells (n = 12) and Müller glia (n = 8) comparing the 10 min before (black) and the 10 min after (white) WIN 55,212-2 application. Only RGCs exhibit a change in CFP/YFP ratio. (**f**) CFP/YFP ratios of RGCs before and after WIN 55,212-2 application using a control $Cl^-$-insensitive M69Q mutant of clomeleon (n = 5). Pre-application of the NKCC1 blocker bumetanide (n = 8), or AM-251 (n = 9) prevents the changes in RGC $Cl^-$ levels. (**g**) With synaptic activity blocked using a cocktail containing DNQX, AP-4, picrotoxin, strychnine, and TTX, the ratio of CFP/YFP fluorescence in RGCs (n = 14), but not amacrine cells (n = 15), still is decreased in response to WIN 55,212-2 application ***p<0.005, two-way RM ANOVA with Holm-Sidak posttest. RGC, Retinal ganglion cell.

To identify RGCs in dissociated retinal cultures, we used Isl2b:EGFP transgenic tadpoles that express GFP in RGCs (*Figure 6g*). Application of WIN 55,212-2 significantly hyperpolarized RGCs (*Figure 6h*, uncorrected for liquid junction potential; n = 6, p=0.004, paired t-test). This shift was prevented in the presence of AM-251 (*Figure 6i*, n = 6, p=0.14, one-way RM ANOVA). This experiment not only confirms that CB1R activation can hyperpolarize RGCs, but the fact that it does so in isolated cells, shows that this effect does not depend on cannabinoid actions on other cells in the retina.

How might CB1R-mediated hyperpolarization result in the enhanced intrinsic excitability of RGCs? RGCs are known to decrease their excitability following a period of depolarization and firing through the inactivation of voltage-gated sodium channels as a way of extending their dynamic range (*Kim and Rieke, 2001*, *2003*; *Weick and Demb, 2011*). Hyperpolarization, by facilitating recovery from inactivation, may therefore increase the intrinsic excitability of RGCs. We examined whether a brief hyperpolarizing pre-pulse could indeed enhance the excitability of Xenopus RGCs in vivo. During whole-cell current-clamp recordings from RGCs, a pre-pulse of 100 ms hyperpolarizing or depolarizing current (range: −100 to +80 pA) was injected, and 25 ms later a 100 ms depolarizing test pulse (200 pA) was applied (*Figure 6j*). The number of spikes evoked by the test pulse was substantially elevated following hyperpolarizing pre-pulses, whereas it decreased after depolarizing pre-pulses (*Figure 6k*). The 25 ms interval between pre-pulse and test pulse facilitated spike counting, but the effect was also evident in experiments where the test pulse followed immediately after the pre-pulse.

## Sustained retinal stimulation induces CB1R-dependent enhancement of RGC firing

eCBs can be released from neurons in response to stimulation driving a prolonged increase in intracellular $Ca^{2+}$ (*Hashimotodani et al., 2007*). We wondered whether in the retina, physiological eCB release might similarly be induced by a sustained increase in luminance. We therefore measured the extracellular RGC response to stimulation of a small central visual subfield (8 degrees) by presenting a 1 s light-OFF test stimulus every 10 s, and attempted to modulate the response by introducing a sustained increase in the luminance of the far surrounding peripheral field for 100 s as the conditioning stimulation (*Figure 7a,b*). In the sample recording traces (*Figure 7b*) and corresponding peristimulus time histogram (*Figure 7c*) shown, conditioning stimulation converted an initially weak set of responses into a robust biphasic response. In a significant majority of cases, sustained peripheral illumination elevated the RGC spike rate evoked by light-OFF test stimuli in the central subfield (*Figure 7e*, before: 4.2 ± 1.6 Hz, after: 8.8 ± 2.4 Hz, n = 10, p=0.042, paired t-test). This conditioning stimulation appeared to recruit responses of RGCs that were not initially activated by the test stimulus (*Figure 7b*) as evidenced by the appearance of new spike forms. Furthermore, the effects of conditioning stimulation were prevented by bath application of AM-251 (before: 6.9 ± 2.8 Hz, after: 4.7 ± 1.6 Hz, n = 10, p=0.244) (*Figure 7b,d,f*), consistent with a CB1R-dependent mechanism.

As predicted by our model, conditioning with peripheral field stimulation also induced a CB1R-dependent decrease of $[Cl^-]_i$ in RGCs. We demonstrated this using in vivo two-photon microscopy to image clomeleon-transfected cells in the eye every 2 min before and immediately after projecting a conditioning peripheral field stimulus, consisting of 100 s of darkness followed by 100 s of peripheral illumination while continuously presenting 1 s flashes in the central subfield every 10 s, through the objective of the two-photon microscope directly onto the retina (*Engert et al., 2002*) (*Figure 7g,h*). In response, we observed a significant reduction in $[Cl^-]_i$ compared with baseline in

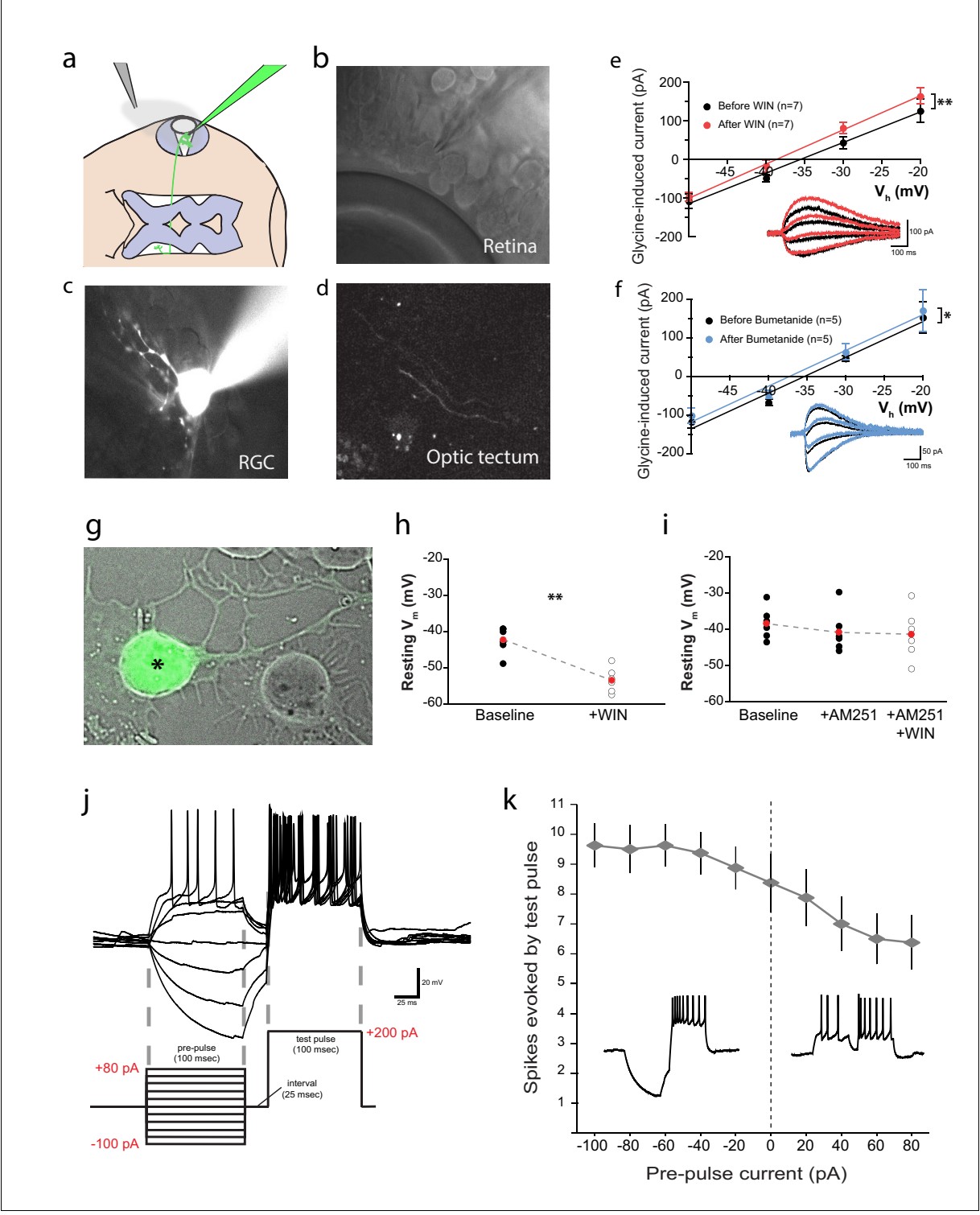

**Figure 6.** CB1R activation hyperpolarizes RGCs by shifting the Cl⁻ equilibrium potential. (a) Schematic of the in vivo whole cell recording set-up to target RGCs for measurement of $E_{Gly}$. (b) Visualized patch clamping of RGC in the eye and (c) confirmatory loading with Alexa Fluor 488 hydrazide. (d) The axon terminal in the tectum of successfully loaded RGC provides confirmation of RGC identity. (e) Following WIN 55,212-2 application, a hyperpolarizing shift in the reversal potential of glycine receptor currents ($E_{Gly}$) is revealed as a leftward shift in I-V curves of peak currents evoked by puffing glycine ($\Delta V = -3.23 \pm 0.84$ mV, n = 7, p < 0.01, two-tailed paired t-test). (f) Bumetanide mimics the shift of $E_{Gly}$ caused by CB1R activation ($\Delta V = -1.60 \pm 0.46$ mV, n = 5, p < 0.05, two-tailed paired t-test). (g) GFP-expressing RGC in dissociated retinal cultures from Isl2b:GFP transgenic tadpoles. (h) Resting membrane potential of RGCs, recorded with gramicidin-perforated patch in dissociated retinal cultures shifted to more hyperpolarized after
*Figure 6 continued on next page*

*Figure 6 continued*

WIN 55,212-2 application (n = 6 cells). (**i**) AM-251 prevented the shift (n = 6 cells; p=0.14, one-way RM-ANOVA). Membrane voltages were not corrected for the liquid junction potential, estimated to be 11.9 mV. (**j**) RGC spiking was evoked by injecting a 200 pA test pulse in current clamp mode 25 ms following pre-pulses ranging from −100 to 80 pA. (**k**) Plot of spike number evoked by the test pulse, as a function of pre-pulse current injected (n = 8 cells) confirms that initial hyperpolarization can result in a greater number of spikes generated in response to identical current injections. Insets are examples of spiking following hyperpolarizing (left) and depolarizing (right) pre-pulses. **p < 0.01, paired t-test. RGC, Retinal ganglion cell.

RGCs (*Figure 7i*, n = 22, p=0.0004, two-way RM ANOVA), and this effect was blocked by pre-treatment with AM-251 (*Figure 7j*, n = 17, p=0.903). In contrast, we did not find any significant reduction in $[Cl^-]_i$ in amacrine (n = 19, p=0.903) or bipolar (n = 10, p=0.831) cells immediately following the conditioning stimulation (*Figure 7k*). Taken together, these data show that CB1R activation can occur naturally in the retina in response to sustained illumination in peripheral visual fields, and it appears to increase the number of RGCs responding to test stimuli.

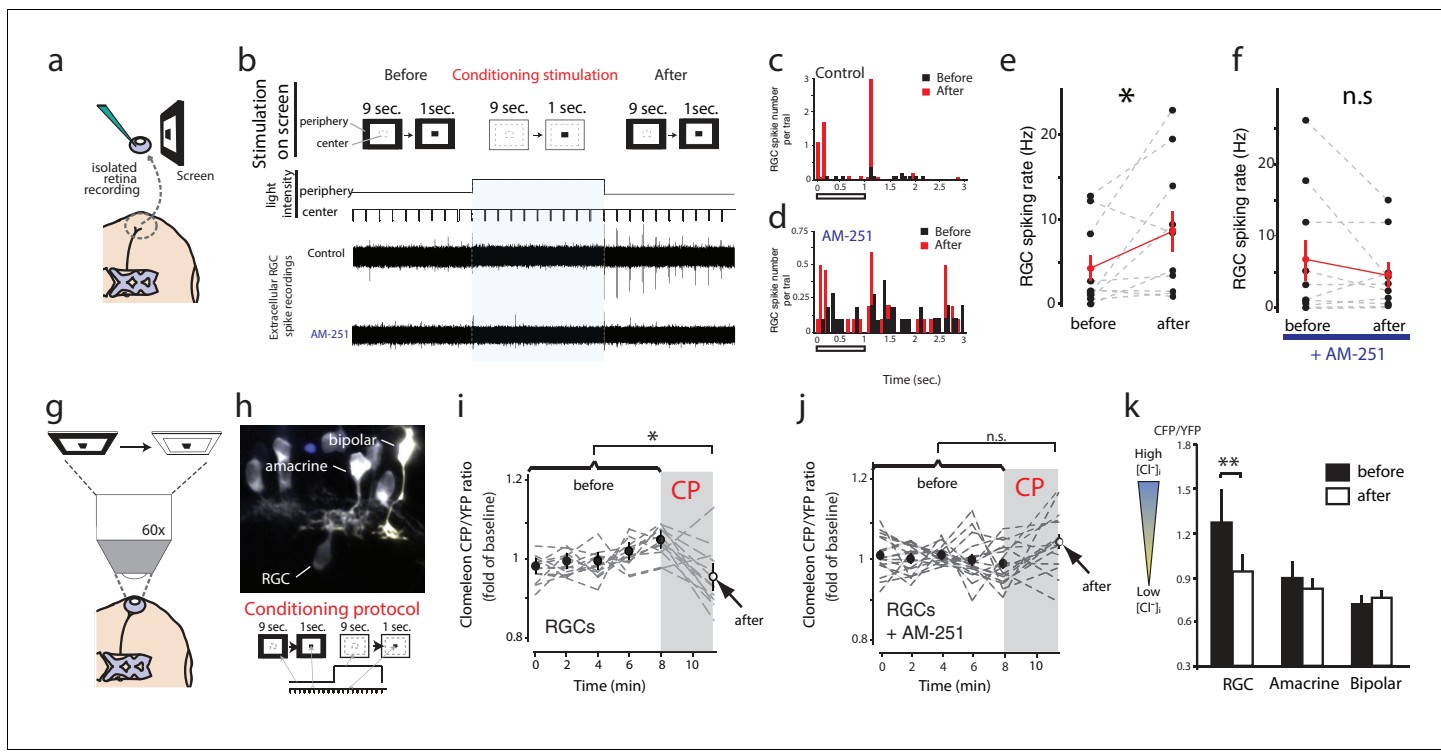

**Figure 7.** Conditioning visual stimulation in peripheral subfields increases RGC spike rates and decreases $[Cl^-]_i$ by a CB1R-dependent mechanism. (**a**) Isolated eye preparation for extracellular retinal recordings, and light stimulation with a changing surround luminance. (**b**) Conditioning visual stimulation protocol displayed on screen. During the initial test period (100 s), 10 light-OFF stimuli (1 s every 10 s) were presented in the central field with a constant dark periphery (Before). This continued during the conditioning period except that periphery visual fields were illuminated continuously for 100 s (conditioning stimulation). After conditioning another set of 10 test stimuli identical to the first set was presented (After). Representative extracellular multiunit RGC spike recordings are shown, in vehicle (Control) and AM-251 (5 μM) containing external solutions. (**c,d**) Binned PSTHs of recorded spikes from the examples illustrated in (b) show that the average response was strongly enhanced after conditioning stimulation in vehicle (**c**) but not with AM-251 treatment (**d**). (**e**) Group data showing the increase in spiking rates of RGCs induced by conditioning stimulation (n = 10 animals). (**f**) This effect is prevented when CB1Rs are blocked with AM-251. (n = 10 animals). (**g–k**) A similar conditioning protocol reduces intracellular Cl⁻ levels selectively in RGCs. Clomeleon signal was imaged in vivo under the two-photon microscope, and the objective was used to project an eyepiece-mounted OLED video display directly onto the imaged region of the retina. (**h**) Example of in vivo clomeleon-expressing retinal cells, and schematic of the visual conditioning protocol (CP). (**i,j**) The decrease in RGC (n = 12 cells) somatic ratios of CFP/YFP after conditioning stimulation (the first time point of the second imaging period compared to the mean of the 5 ratio values during the first imaging period) was blocked in AM-251 (n = 17). (**k**) Only RGCs (n = 22), but not amacrine cells (n = 19) or bipolar cells (n = 10) exhibited this decrease in intracellular Cl⁻ after conditioning stimulation. (**e, f**) *p<0.05, paired t-test, (**k**) **p<0.01, two-way RM ANOVA with Holm-Sidak posttest. RGC, Retinal ganglion cell.

## Cannabinoid modulation of RGC firing impacts response properties in higher visual areas

Having described a mechanism by which CB1R activation in the retina increases intrinsic excitability of RGCs and mediates certain forms of visual response facilitation in the retina, we next assessed the potential consequences of this phenomenon on visual processing in the brain. We first performed amphotericin-perforated patch recordings of visually evoked synaptic potentials from neurons in the optic tectum which receives direct RGC innervation – in this case, perforated-patch recording helps maintain the quality of recordings during long recordings by preventing 'wash-out' of the cell. Direct electrical stimulation of RGCs through a glass-stimulating electrode placed in the ganglion cell layer of the eye (*Figure 8a*) evoked α-amino-3-hydroxy-5-methyl-4-isoxazolepropionic acid (AMPA) receptor-mediated retinotectal currents with latencies of about 5 ms in postsynaptic tectal neurons clamped at −60 mV. Following bath application (perfusion over the entire tadpole) of WIN 55,212-2 (1 μM), peak amplitudes of retinal evoked EPSCs were significantly increased (*Figure 8b,c*; 151.9 ± 17.5% baseline, n = 11, p=0.0004). This increase in EPSC amplitude was dependent on CB1R activation, as it was blocked by the inverse agonist AM-251 (*Figure 8c*, n = 5, p=0.497). Increasing the endogenous AEA concentration with URB597 similarly increased EPSC amplitude (*Figure 8c*, n = 7, p=0.029). Treatments with either N-arachidonyl maleimide (NAM, 150 nM, n = 7, p=0.988) or JZL184 (100 nM, n = 7, p=0.999), two selective inhibitors of monoacylglycerol lipase (MAGL), the 2-AG degrading enzyme, showed no enhancement (*Figure 8—figure supplement 1a*), providing further evidence that AEA is likely to be the endogenous cannabinoid that mediates the augmentation of the response to retinal stimulation. The WIN 55,212-2-induced enhancement of retinotectal EPSC amplitudes were blocked by strychnine and bumetanide (*Figure 8—figure supplement 1b*), mirroring their effects on RGCs recorded in the isolated retinal eyecup (*Figure 4e–h*) and suggesting that the changes in RGC excitability probably underlie the enhanced retinotectal responses to stimulation in the eye.

To determine whether CB1R agonist perfusion was enhancing responses through action in the eye or in the tectum, we used an isolated brain preparation in which the eyes were no longer present and instead stimulated the severed RGC axons in the optic tract directly using a bipolar-stimulating electrode (*Wu et al., 1996*). The difference in the peak amplitude before and after WIN 55,212-2 application could no longer be detected (*Figure 8d*, 100 ± 11% baseline, n = 7, p=0.999). This indicates that the cannabinoid-mediated enhancement of retinotectal EPSC amplitude is due to increased retinal excitability and the resulting recruitment of more RGCs when stimulating in the eye, rather than to changes at the retinotectal synapses or in properties of the neurons within the optic tectum (*Figure 8—figure supplement 1c–h*).

Taken together, these data suggest a model in which activation of CB1Rs on RGCs should enhance their excitability enough to recruit RGCs to fire action potentials in response to visual stimulation of subfields that were previously sub-threshold. This should manifest as an enlargement of input receptive field (RF) sizes in tectal neurons, which receive convergent inputs from many RGCs. To test this prediction, we examined whether RFs of tectal neurons were expanded by WIN 55,212-2 treatment. Immediately after establishing stable in vivoamphotericin-perforated patches onto tectal neurons in intact tadpoles, the objective of the electrophysiology microscope was reoriented to project a 7 × 7 grid of rectangular subfields directly onto the photoreceptor layer of the eye contralateral to the cell being recorded. Each subfield was then repeatedly stimulated in random sequence with a light-OFF stimulus while recording compound synaptic currents from tectal cells held at −60 mV (*Engert et al., 2002*; *Tao and Poo, 2005*) (*Figure 8e*). An example of one RF mapping experiment is shown in *Figure 8f–g*. WIN 55,212-2 application produced a significant increase in tectal RF size (127.6 ± 9.4% baseline, n = 5, p=0.042, one-sample t-test) and in the total charge evoked (140.3 ± 14.2% baseline, n = 5, p=0.047) across all responsive subfields (*Figure 8f–h*). These results show that activation of retinal CB1Rs increases the size and strength of tectal RFs, most likely through the recruitment of responses from initially subthreshold RGC inputs.

## CB1R activation impacts visually driven behavior

To determine whether the enhancement of visual responses by CB1R activation could contribute meaningfully to vision, we exploited the instinctive avoidance of dark moving dots by freely swimming tadpoles (*Dong et al., 2009*). We created video-tracking software to automatically track

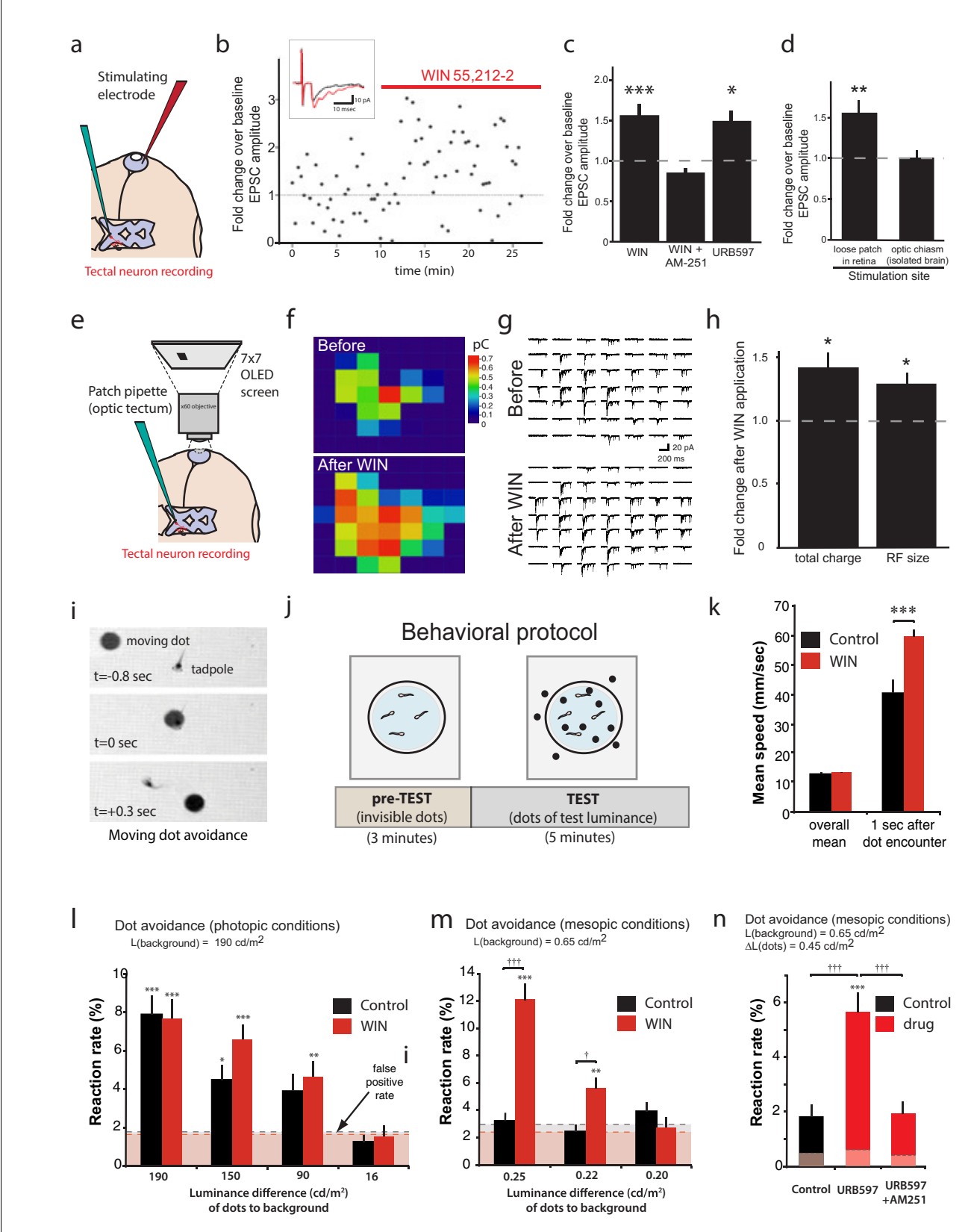

**Figure 8.** CB1R activation increases retinal inputs onto tectal neurons, expanding their input RFs, and enhances performance in a visual detection task. (**a**) Schematic of in vivo amphotericin-perforated patch recordings of tectal neurons with extracellular electrical stimulation in the retina. (**b**) Peak

*Figure 8 continued on next page*

*Figure 8 continued*

retinotectal EPSC amplitudes show an increase induced during perfusion of WIN 55,212-2 onto the tadpole (red line) in a representative experiment. Inset illustrates recording traces before (black) and after (red) drug. (c) The increase in retinotectal EPSC amplitude by WIN 55,212-2 (n = 11) was blocked by AM-251 (n = 5). Elevating endogenous AEA using URB597 (n = 7) similarly enhanced EPSC amplitudes. (d) An isolated brain preparation (n = 7) in which the eyes are not present and retinotectal axons are stimulated directly at the chiasm did not show EPSC enhancement following WIN 55,212-2 application, indicating a retinal site of action. (e) Tectal RF mapping was performed by projecting visual stimuli from a small OLED video monitor onto the retina through the microscope objective while recording from tectal neurons. (f) Example color coded RF maps of integrated postsynaptic charge evoked by OFF stimuli presented in a 7 × 7 grid of visual fields before and after WIN 55,212-2 application reveal an expansion of the RF. (g) Sample traces of the compound synaptic currents used to generate f. (h) Total postsynaptic charge evoked by stimulating at each field of the grid, and the mean sizes of tectal cell RFs both increased in response to WIN 55,212-2 application (n = 5 animals). (i–n) Visually guided escape behavior analysis (i) Avoidance response of a free swimming tadpole to a dark moving dot. (j) False-positive rate was measured during a 3 min pre-TEST where dots possess the same luminance as the background ('invisible dots'); this was followed by a 5 min TEST period with dots darker than the background. (k) WIN 55,212-2-treated tadpoles swam at the same speed as control tadpoles prior to encountering dots, but swam away faster following an encounter. (l) The probabilities of eliciting avoidance reactions to dots presented at a range of contrasts on a bright background (luminance: 190 cd m$^{-2}$) did not differ between control (n = 24) and WIN 55,212-2-treated (n = 24) tadpoles. (m) Dark dots moving on a dim background (luminance: 0.65 cd m$^{-2}$) led to avoidance reactions in WIN 55,212-2-treated (n = 24), but not control (n = 24), tadpoles. (n) URB597-treated tadpoles (n = 24) also reacted with higher rates than matched control animals (n = 24). This was prevented by blocking CB1Rs with AM-251 (n = 24). Reaction rates represent the average fraction of encounters with dots per tadpole that resulted in a change in swimming velocity (>24 mm s$^{-1}$) In l, m and n horizontal dashed lines represent mean false-positive reaction rates measured pre-TEST using 'invisible dots'. (c,d,k) *p<0.05, **p<0.01, two-way RM ANOVA with Holm-Sidak posttest. (h) *p<0.05, one-sample t-test vs. pre-drug baseline. (l-n) †††p<0.001 two-way ANOVA with Holm-Sidak post-hoc tests comparing control and drug-treated groups, ***p<0.001; **p<0.01, *p<0.05 compared to pre-TEST false-positive rates. AEA, anandamide; EPSC, Excitatory postsynaptic current; RF, Receptive field.

The following figure supplement is available for figure 8:

**Figure supplement 1.** Effects of CB1R activation on tectal neurons.

tadpole and dot movements and quantify escape responses over a range of brightness and contrast conditions in control or drug-containing media. Videos of up to 8 animals at a time, swimming in a Petri dish positioned over a computer monitor were collected for each trial (*Figure 8i,j*, *Video 1*). A potential dot avoidance event was counted whenever a moving dot came within 5 mm of a tadpole. A successful avoidance event was logged whenever swimming velocity changed by at least 24 mm s$^{-1}$ within 0.25 s of initial encounter. During an initial 3 min pre-test period, a uniform field was displayed to calculate false-positive response probabilities for the tadpoles by measuring escape 'responses' to 'invisible dots' presented by the software as isoluminant to background, and during the subsequent 5 min test period, dots (4 mm diameter) with lower luminance than the background, moving in random directions were displayed on the monitor while the software tracked each tadpole for escape responses (*Figure 8j*). Importantly, WIN 55,212-2 treatment did not impair basic motor coordination. The mean velocities of swimming tadpoles when not interacting with dots was indistinguishable between control and WIN 55,212-2-treated groups, indicating that the drug did not alter the rhythms or coordination underlying swimming behavior (*Figure 8k*, n = 24, p=0.491, two-way RM ANOVA with Holm-Sidak posttest). However, we did find that WIN 55,212-2-treated tadpoles swam significantly faster when executing escape behaviors (n = 24, p<0.001), highlighting the caveat that higher-order circuitry was likely also affected by the drug treatments.

To compare visual contrast sensitivity of control tadpoles versus tadpoles treated with WIN 55,212-2, we measured their relative frequencies of dot avoidance reactions to dots of different luminances relative to background. When dark dots were presented under photopic conditions (overall luminance of the field = 190 cd m$^{-2}$), control tadpoles and WIN 55,212-2-treated tadpoles did not differ in their avoidance response rates to dots having a luminance 190, 150 and 90 cd m$^{-2}$ darker than background. All elicited

**Video 1.** Control tadpole response to dots under photopic illumination.

significantly more escapes than the false-positive rate measured during the pre-test periods. The faintest dots, which differed from background by only 16 cd m$^{-2}$, did not elicit a significant reaction (*Figure 8l*). Under mesopic light conditions (overall luminance of the field = 0.65 cd m$^{-2}$) control animals responded only at chance levels to all stimuli tested. Remarkably, WIN 55,212-2-treated tadpoles successfully avoided a significant fraction of dots with luminance differences to the background as small as 0.22 cd m$^{-2}$ (*Figure 8m*, n = 24, p=0.003, *Videos 2* and *3*). Furthermore, tadpoles treated with URB597 to elevate endogenous cannabinoid levels also showed significantly enhanced dot avoidance compared to untreated controls (*Figure 8n*, n = 24, p<0.001). The effects of URB597 were abolished by the CB1R inverse agonist AM-251 (5 µM), indicating that the drug was indeed acting by elevating endogenous cannabinoids (n = 24, p<0.001).

These results demonstrate that endogenous AEA acting on CB1Rs is able to enhance visual contrast sensitivity. Although our observation of increased escape velocities suggests that the effects of WIN 55,212-2 were not limited to the retina, the finding that drug-treated tadpoles exhibited superior detection of mesopic, but not photopic, visual stimuli, strongly argues for a retinal site of action. It is therefore highly likely that the enhanced excitability of RGCs by CB1R activation which we first report here, also contributes to this remarkable improvement in the visual detection of dim objects.

## Discussion

We found that activation of CB1Rs on RGCs reduces [Cl$^{-}$]$_i$, through the AMPK-dependent inhibition of NKCC1. This results in the hyperpolarization of the RGC resting membrane potential by enhancing the driving force for Cl$^{-}$ currents through GlyRs, and acutely enhances the intrinsic excitability of the RGCs, most likely through the resultant de-inactivation of voltage-gated sodium channels. This enhancement in RGC intrinsic excitability leads to the recruitment of additional, initially subthreshold RGC responses by target neurons in the optic tectum, which in turn may contribute to improved visual contrast sensitivity.

CB1Rs constitute one of the most highly expressed G-protein-coupled receptors in the central nervous system (*Herkenham et al., 1990*), where they generally have been reported to act to reduce neurotransmitter release by attenuating voltage-dependent Ca$^{2+}$ or K$^{+}$ conductances in presynaptic terminals (*Chevaleyre et al., 2006*; *Kano et al., 2009*; *Piomelli, 2003*). CB1R-mediated hyperpolarization has also been proposed to reduce intrinsic excitability in cerebellar and cortical interneurons by positive modulation of a K$^{+}$ conductance (*Bacci et al., 2004*; *Kreitzer et al., 2002*). Our work presents a novel mechanism for interaction between glycinergic inhibition and the eCB system involving the regulation of Cl$^{-}$ homeostasis, to enhance intrinsic neuronal excitability.

This finding also importantly adds to our understanding of the role of chloride transporter regulation in neuronal physiology. In neurons, Cl$^{-}$ levels are primarily controlled by the interplay of the inward NKCC1 and outward KCC2 cation-Cl$^{-}$ co-transporters, which are modulated by phosphorylation (*Darman et al., 2001*; *de Los Heros et al., 2006*; *Delpire and Austin, 2010*; *Garzón-Muvdi et al., 2007*; *Rinehart et al., 2009*; *Song et al., 2002*; *Wenz et al., 2009*). Activation of human NKCC1 requires phosphorylation at T217 by SPAK family members downstream of WNK1 (*Darman et al., 2001*; *Vitari et al., 2006*). It had previously been reported that AMPK activation can result in the inhibition of human NKCC1 through phosphorylation at S77, as well as suppression of T217 phosphorylation by SPAK (*Fraser et al., 2014*). In our study, we have provided evidence that

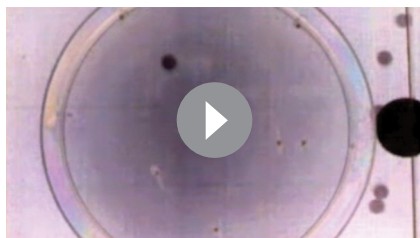

**Video 2.** Control tadpole response to dots under mesopic illumination.

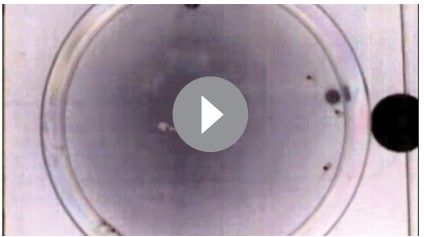

**Video 3.** WIN 55,212-2-treated tadpole response to dots under mesopic illumination.

CB1R-mediated down-regulation of $[Cl^-]_i$. in RGCs is dependent upon the AMPK pathway.

We used Clomeleon imaging and a variety of electrophysiological methods to demonstrate the reduction of $[Cl^-]_i$ at the cell soma, and further found that $Cl^-$ currents through GlyRs appear to participate in the regulation of resting membrane voltage and cell excitability, presumably through tonic inhibition. Although tonic glycinergic inhibition has been well studied in the spinal cord and hippocampus (*Xu and Gong, 2010*), our study is one of the first to suggest a role for tonic inhibition by strychnine-sensitive GlyRs in the retina. We also found evidence from calcium imaging of bipolar cell terminals for their regulation by cannabinoid signaling (*Figure 3b–e*) and observed that synaptic excitation and inhibition evoked by visual stimuli are reduced in response to CB1R activation (*Figure 3m–p*). While a reduction in synaptic inhibition could potentially contribute to the cannabinoid-dependent enhancement of RGC firing that we detected, it is unlikely to fully account for this phenomenon for three reasons. First, the primary effect that we observed was a specific decrease in RGC $[Cl]_i$ leading to membrane hyperpolarization. This should enhance inhibitory driving force rather than decrease it and therefore is more consistent with our model that tonic pre-hyperpolarization helps prime the cell to fire more robustly to subsequent depolarization, which we demonstrated can occur in RGCs (*Figure 6j,k*). Second, the fact that direct, instantaneous electrical stimulation of RGCs with an electrode in the RGC layer, which entirely bypasses upstream retinal circuitry, still resulted in a strychnine-sensitive enhancement of retinotectal EPSCs by CB1R activation is strongly indicative of direct tonic changes in RGC excitability. Third, when measuring evoked firing rates of RGCs, we did not find that pharmacological blockade of GABA-A receptors or GlyRs increased the excitability of RGCs. To the contrary, the fact that inhibition of NKCC1 or blockade of GlyRs completely eliminated the CB1R-dependent enhancement of RGC firing rates, taken together with Clomeleon imaging data showing that $[Cl^-]_i$ is selectively altered in RGCs argues that direct, cell-autonomous changes in RGC excitability rather than modulation of upstream synaptic transmission constituted the primary mechanism underlying the increased firing in response to visual stimulation. The CB1R-mediated reduction of $Ca^{2+}$ in bipolar cell terminals seems more likely to impact vision by attenuating the signal from the outer retina to RGCs as has been well-documented in other species (*Middleton and Protti, 2011*; *Straiker et al., 1999*; *Yazulla et al., 2000*).

Additionally, GlyRs have been reported to be directly positively modulated by cannabinoids (*Lozovaya et al., 2005*). The change in slope of the $E_{Gly}$ I-V curve upon WIN 55,212-2 perfusion, apparent in *Figure 6e*, may provide further support for this idea. GlyR modulation could act synergistically with the augmentation of tonic inhibition via the changes in $[Cl^-]_i$ that we demonstrated to enhance intrinsic excitability of RGCs. On the other hand, it would counter any mechanism that relied on synaptic disinhibition. A direct enhancement of GlyRs by cannabinoids might account for why strychnine, but not gabazine, was most effective at preventing the effects of WIN 55,212-2 on RGC excitability.

Cannabinoid regulation of RGC-firing properties is not entirely unprecedented. Suppression of HVA $Ca^{2+}$ currents has been reported in dissociated rat RGCs (*Lalonde et al., 2006*). Interestingly, RGCs are known to express both the synthetic (NAPE-PLD) and degradative (FAAH) enzymes for AEA (*Maione et al., 2009*; *Yazulla et al., 1999*), suggesting they also may be a source of eCBs. In support of this idea, the CB1R-dependent reduction in synaptic drive from bipolar and amacrine interneurons can be induced by direct RGC membrane depolarization (*Wang et al., 2016*). Based on our present data, we propose that AEA release additionally acts in an auto- or paracrine manner to enhance RGC excitability. Opposing actions on bipolar cell terminals and RGCs may serve a useful role to improve the signal-to-noise ratio of visual responses by maintaining a vigorous retinal output while attenuating excessive synaptic noise from the outer retina (*Grimes et al., 2014*). This may explain our finding that CB1R activation dramatically improved behavioral responses to dark moving cues near absolute threshold for visual detection.

Finally, our work describes a mechanism of sensitization to spatial subfield stimulation when a conditioning stimulus is presented in the periphery. This is reminiscent of the so-called 'Westheimer effect' where detection sensitivity for a small test flash at the center of a background disk improves as the background disk increases in size, an effect shown in humans (*Westheimer, 1965*) and lower vertebrates (*Burkhardt, 1974*). The sensitization component of this effect generally has been explained based on the antagonistic center and surround regions of the RGC RF, such that progressive stimulation in the surround decreases net excitation. However, important limitations to this model exist (*Essock et al., 1985*; *Sinai et al., 1999*) and recent research has revealed that

sensitization to temporal contrast changes can last for seconds in individual regions of the RGC surround field and may therefore mediate a predictive function (*Kastner and Baccus, 2013*). The CB1R-dependent sensitization of mesopic RGC responses that we report here can persist for minutes, but appears to require prolonged stimulation in the periphery to produce the eCBs that induce it. This mechanism may be beneficial as a means of optimizing RGC signal-to-noise ratios during the initial period of dark adaptation before photoreceptors obtain maximum sensitivity.

The CB1R-dependent mechanism for low luminance sensitization that we describe here in the developing amphibian retina will require additional investigation to determine whether it also occurs in the mammalian eye, although the psychophysical evidence is encouraging (*Russo et al., 2004*; *West, 1991*). Our findings offer a new framework for future studies of contrast gain control in the retina and generally expand our understanding of the diverse mechanisms of neuromodulation by eCBs.

## Materials and methods

### Animals

Albino *Xenopus laevis* tadpoles (RRID:NXR_0.0082) were bred by human chorionic gonadotropin-induced mating. Embryos were reared in a biological oxygen demand incubator (VWR) at 21C in standard modified Barth's saline with HEPES (MBSH) in a 12 hr:12 hr light-dark cycle. Experiments were performed during the light period. All experiments were approved by the Montreal Neurological Institute Animal Care Committee in accordance with Canadian Council on Animal Care guidelines. Tadpoles were developmentally staged according to the standard criteria of Nieuwkoop and Faber (*Nieuwkoop and Faber, 1994*).

### Immunohistochemistry

For cryostat sections, stage 42–45 tadpoles were fixed in 4% paraformaldehyde in 0.1 M phosphate buffer and then cryoprotected overnight at 4C in 30% sucrose (Sigma, St. Louis, MO), after which they were transferred to Tissue-Tek O.C.T. Compound (PELCO International, Redding, CA) for 20 μm sectioning. Sections were permeabilized (0.3% triton X-100, Sigma) and incubated in blocking solution containing 5% normal goat serum (Gibco, Grand Island, NY), 10% bovine serum albumin and 0.3% triton X-100 for 1 hr followed by application of primary antibody for 2 hr. CB1Rs were labeled using a rabbit antibody (RRID:AB_2314112) against the N terminal of the CBR1R (1/100, a generous gift of Dr. Ken Mackie) (*Cesa et al., 2001*), RGCs were electroporated with EGFP and labeled using a chicken polyclonal GFP antibody (RRID:AB_300798, 1/500, Abcam, Cambridge, MA). Secondary antibodies were conjugated to Alexa Fluor 488 (RRID:AB_142924) or 555 (RRID:AB_2535849) (Invitrogen) and applied for 1 hr. For all immunolabeling, control experiments omitting the primary antibody were performed. We also controlled for the specificity of the primary antibody by co-application of a blocking peptide provided by Dr. Ken Mackie. Slides were then rinsed in phosphate buffer and coverslipped in Vectashield Mounting Medium with DAPI nuclear stain (Vector Laboratories, Burlingame, CA). Sections were imaged on a Zeiss LSM 710 inverted confocal microscope.

### Drugs

The following drugs were used in the experiments reported here: WIN55,212–2 (1 μM), ACEA, (1 μM) AM251 (5 μM), URB597 (2 μM), JZL184 (100 nM), N-Arachidonyl Maleimide (NAM, 150 nM), from Cayman Chemicals (Ann Arbor, MI); Strychnine HCl (60 μM), and Bumetanide (10 μM) from Sigma Aldrich; Gabazine (SR95531; 6 μM), D-APV (50 μM), L-AP4 (50 μM), DNQX (20 μM), picrotoxin (100 μM) and glycine from Tocris (Minneapolis, MN); and TTX (1 μM) from Alomone Labs (Israel).

### Electroporation

Cells in the retina were bulk electroporated as described (*Ruthazer et al., 2006*). Cells were imaged roughly 48 hr after electroporation. In brief, glass micropipettes were made from borosilicate capillaries pulled on a PC-10 puller (Narishige, East Meadow, NY). Pipette were loaded with plasmid solution and attached to a custom-made pressure injection system. Plasmid solution (0.5–5 μg/μL) was then pressure-injected in the eye, without visibly distending the eye, and current was delivered across custom-made platinum plate electrodes placed on either side of the eye using 3 pulses (36

Volts, 1.6 ms) in each polarity using a constant voltage stimulator (Grass SD-9) with a 3 µF capacitor placed in parallel.

## In vivo imaging of GCaMP6s

Stage-40 tadpoles were electroporated in the retina with plasmid DNA encoding GCaMP6s and given at least 24 hr to express the protein. Stage 42–45 tadpoles were immobilized by bath application of pancuronium bromide (2 mM, Sigma) and placed in a custom-made imaging chamber, eye facing up with the surrounding skin gently nicked for drug permeability. Tadpoles expressing the construct in bipolar cells were selected for further in vivo two-photon microscopy using a Thorlabs multiphoton imaging system with resonant scanner and Olympus 20x 1.0 NA immersion objective (Thorlabs, Newton, NJ). A MaiTai-BB Ti:sapphire femtosecond pulsed laser (Spectra Physics, Santa Clara, CA)set at 910 nm was used for fluorescence excitation. For visual stimulation, an A310 Accu-pulser (WPI, Sarasota, FL) was used to drive a red LED (627 nm, rebel red Luxeon Star) to present trains of light flashes (ON for 1 s, every 10 s). To visualize GCaMP6s fluorescence intensity changes together with light stimulation, XYT series images of the retina were acquired at 10 Hz for 2 min before and 5 min after WIN-55,212-2 (1 µM) application. Images were acquired on a PC using Thor-Image LS software and subsequently analyzed using ImageJ (NIH). Regions of interest (ROI) were drawn manually around visually identifiable bipolar terminals, and the mean intensity of the ROIs in the green channel was determined for each frame of the XZT series. The background intensity was measuring by calculating the mean intensity of a large ROI in an area free of any fluorescence, and subtracted for each frame from the intensity of the ROI of bipolar cell terminals. $F_0$ was calculated as an average of the (ROI intensity – background intensity) for the initial 20 frames before the beginning of light stimulation. The change in fluorescence was reported as $\triangle F/F_0$, where $\triangle F = F_{(t)} – F_0$.

## In vivo imaging of clomeleon

Stage-40 tadpoles were retinally electroporated with a construct encoding a modified Clomeleon and given at least 24 hr to express the protein. Stage 42–45 tadpoles were immobilized by a pressure injection of tubocurarine (1 µM) in the tail and placed in a custom-made imaging chamber, eye facing up with the surrounding skin gently nicked to ensure drug permeability. In vivo two-photon microscopy of retinal cells was performed using a Olympus FV300 confocal microscope custom-converted for multiphoton imaging equipped with a MaiTai-BB Ti:sapphire femtosecond pulsed laser set to excite at 860 nm. To visualize clomeleon fluorescence intensity changes, XYZT series images of the retina were acquired simultaneously on CFP and YFP channels every 2 min. After a first 10 min of baseline imaging, another 10 min of imaging followed, during which various drugs were applied. Images were acquired on a PC using Fluoview software and subsequently analyzed using Image J (NIH). Bleed-through correction (PixFRET plugin [*Feige et al., 2005*]) from the CFP signal to the YFP channel, and background subtraction were performed prior to calculating cyan:yellow intensity ratios. Contol experiments using a chloride-insensitive mutant Clomeleon-M69Q demonstrated that the fluorescence intensity changes indeed reflected $[Cl^-]_i$ rather than any potential residual pH sensitivity of the YFP.

## Electroretinogram recordings

Stage 42–45 tadpoles were anesthetized in MBSH containing 0.02% MS222 (Sigma) for dissection. The eye was removed using a 30 ga needle. For recording, the eye was gently laid on top of a filter paper supplied with a constant flow of fresh external solution at room temperature. External solution consisted of (in mM): NaCl, 115; KCl, 2; HEPES, 10; CaCl2, 3; MgCl2, 1.5; glucose, 10; glycine, 0.005 (pH 7.3). ERGs were recorded by placing a borosilicate glass patch electrode (O.D.:1.5 mm, O.D.: 0.86 mm, 10 cm length, Sutter Instrument) filled with 1 M NaCl with a tip diameter of 2–6 µm under the lens on the surface of the retina using an anterior transscleral approach. A stable ERG recording was typically achieved immediately. The isolated eye was placed with the cornea facing up few centimeters away from a full-field LED screen light source (light-ON stimulation = 10 s OFF followed by 0.5 s ON; light-OFF stimulation = 10 s ON followed by 0.5 s OFF). ERGs were amplified using an extracellular amplifier (A-M Systems 1800, Sequim, WA) set to 1000x gain, and low-pass filtered at 300 Hz. The ERG traces and action potentials were analyzed by using custom-written software in IGOR Pro (WaveMetrics). ERG recordings were verified by application of L-AP4 (50 µM), which

selectively blocks transmission between photoreceptors and ON-bipolar cells in the retina, and fully suppressed the b-wave as expected (*Figure 2—figure supplement 1*).

## Ganglion cell recordings

Recordings of visually evoked spiking in the retina used the same preparation as described for ERG recordings. Because RGCs are the dominant spiking cells of the retina (along with AII amacrine cells) (*Boos et al., 1993*), we consider our spike recordings to reflect mainly RGC activity. After a cell was found, a period of 10 min was given to confirm the stability of the recording. The amplifier was set to filter 300 Hz–1 kHz. The spike traces were analyzed using custom-written software in IGOR Pro. To drive RGCs spiking, we either used full-field flash, as described in the ERG recordings section, or a complex stimulation designed to sensitize RGC responses through a light conditioning protocol.

For this stimulation, we displayed subfield OFF-flash in the center (1 s. light-OFF flash every 10 s., light OFF luminance of $0.61 \pm 0.09$ cd.m$^{-2}$) with a steady ON- or OFF-surround (100 s. OFF, followed by 100 s. ON, and back to 100 s. OFF). The light ON luminance was $15.56 \pm 2.80$ cd.m$^{-2}$. We compared the spiking responses between the first and second OFF-surround periods, evoked by 10 successive OFF-centered flashes, and described the central period of ON-surround as the conditioning part of the light stimulation protocol.

For whole-cell RGC recording of synaptic currents, the recording pipettes were filled with an internal solution containing (in mM): K-gluconate, 100; KCl, 8 NaCl, 5; MgCl2, 1.5; EGTA, 0.5; HEPES, 20; ATP, 2; GTP, 0.3 (mOsm 255; pH 7.3). External solution contained (in mM): NaCl, 115; KCl, 4; HEPES, 5; CaCl2, 3; MgCl2, 3; glucose, 10 (mOsm 255, pH 7.3). To drive the responses to light in voltage clamp mode, RGCs were held successively at $-65$ mV (to record EPSCs) and 0 mV (to record IPSCs), and stimulated using a red LED (627 nm, rebel red Luxeon Star) to present trains of light flashes (ON for 3 s, every 8 s). Each measurement was made 10 times per cell.

$E_{Gly}$ was calculated from peak amplitudes of glycine-induced IPSCs recorded at different command potentials (from $-80$ to $+10$ mV with step increments of $+10$ mV) as an intersection between the x-axis and the fit of the observed amplitudes. Glycine (500 μM) was puffed every 30 s (30 psi for 5–20 ms) through a patch pipette connected to a Picospritzer (Parker Instrumentation, Canada). For each cell, the $E_{Gly}$ measurement protocol was repeated twice before and after drug application.

## Dissociated retinal cell culture

For each culture, retinae were dissected in disaggregation solution (115 mM NaCl, 4 mM KCl, 5 mM HEPES, 10 mM glucose, pH 7.3, osmolarity at 250 mOsmol) containing 0.02% MS222 anesthetic, and then incubated for 8 min at room temperature in a papain solution (10 U/mL) in disaggregation solution, containing 0.4% DNase, and L-cystein HCl (1.26 mM). Retinae were then triturated in LO-Ovomucoid solution (115 mM NaCl, 4 mM KCl, 5 mM HEPES, 10 mM glucose, 3 mM CaCl2, 3 mM MgCl2, 10% BSA, 1.15 mg/mL trypsin inhibitor from soybean, pH 7.3), and spun 11 min at 200 ×g at room temperature. The supernatant was removed and retinae were resuspended in reaggregation solution (115 mM NaCl, 4 mM KCl, 5 mM HEPES, 10 mM glucose, 3 mM CaCl2, 3 mM MgCl2, pH 7.3), and seeded on poly-D-lysine (>300,000 kDa) treated coverslips (10 retinae/coverslip).

## Gramicidin perforated patch recordings from cultured RGCs

After 18–24 hr in culture, dissociated retinal cells (as described above) from the Isl2b:GFP line (*Boos et al., 1993*) were gently transferred to the recording chamber, which was constantly perfused at low rate with fresh external solution containing (in mM): NaCl, 115; KCl, 4; HEPES, 5; CaCl2, 3; MgCl2, 3; glucose, 10 (mOsm 255, pH 7.3). Only GFP-positive cells that were well attached to the coverslip and that had already grown several neurites were used for recordings. Gramicidin-perforated patch recordings were performed as previously described (*Khakhalin and Aizenman, 2012*). Borosilicate glass micropipettes (Sutter Instruments), with a resistance of 8–12 MOhms were briefly dipped in gramicidin-free internal solution, and then back-filled with internal solution containing (in mM): Kgluconate, 100; KCl, 8; NaCl, 5; MgCl2, 1.5; EGTA, 10; HEPES, 20; ATP, 2; GTP, 0.3 (mOsm 255; pH 7.3) supplemented with gramicidin 20 μg/ml (Sigma). To prevent gramicidin spillover, no positive pressure was applied to the tip of the pipette while approaching the cells. Recordings were

usually possible 10–20 min after successful formation of the gigaseal. Membrane potentials were not corrected for liquid junction potentials.

## In vivo tectal cell recordings

Stage 42–45 tadpoles were anesthetized in MBSH containing 0.02% MS222 (Sigma) for dissection. The lens was gently removed from one eye to expose the retina. The skin on the head was cut, and the brain was opened along the midline for tectal recording. For recordings, the tadpole was fixed to a PDMS insert in the recording chamber with insect pins. The tadpole was constantly perfused with fresh external solution, and all experiments were performed at the room temperature. External solution consisted of (in mM): NaCl, 115; KCl, 2; HEPES, 10; CaCl2, 3; MgCl2, 1.5; glucose, 10; glycine, 0.005 (pH 7.3). Amphotericin B perforated-patch recording, which avoids rundown sometimes encountered with whole-cell recording, was performed as previously described (*Tsui et al., 2010*; *Zhang et al., 2000*). Borosilicate glass micropipettes (Warner, Hamden, CT), with a resistance in the range of 4–7 MOhm were briefly dipped in internal solution, and then back-filled with internal solution containing amphotericin B (250 µg ml$^{-1}$). The internal solution contained (in mM): Kgluconate, 110; KCl, 10; NaCl, 5; MgCl2, 1.5; EGTA, 0.5; HEPES, 20; ATP, 2; GTP, 0.3 (pH 7.3). The same size patch pipette was used for extracellular RGC stimulation except that it was filled with external solution. For optic chiasm stimulation experiments, the brain was dissected out into external solution, filleted flat by transecting the dorsal midline and pinned to a Sylgard base in a recording perfusion chamber. A custom-bent 25 mm cluster electrode (FHC, Maine) was positioned onto the optic chiasm through which constant current electrical stimulation was delivered using a stimulus isolation unit (AMPI, Israel). Test pulses to evoke retinotectal EPSCs were applied every 30 s in voltage clamp. Recordings were acquired with a patch-clamp amplifier (Axopatch 200B; Molecular Devices, Sunnyvale, CA) and Clampex software (Molecular Devices). Signals were filtered at 2 kHz and sampled at 5 kHz. Input resistance (0.5–1 GOhm) and series resistance (30–70 MOhm) were monitored continuously during recordings. Data were accepted for analysis only if the series resistance and input resistance remained relatively constant (<20% change) throughout the experiment. Cells were held at a holding potential of −60 mV. To measure visual receptive fields of tectal neurons, visual stimuli were presented on an 800 × 600 pixel, 9 × 12 mm SVGA 3D OLED-XL color microdisplay (eMagin, Hopewell Junction, NY) mounted in place of one microscope eyepiece using a custom built frame. The image was then focused onto the photoreceptor layer of the retina through the microscope objective. The screen displayed a 7 × 7 pixel array, each pixel (30 µm square) of which independently turned off for 200 ms in random order at 0.067 Hz to drive receptive field OFF stimulation. Total charge transferred in the 200 ms window following each stimulus presentation was used to calculate input strength for the corresponding part of the visual field.

## Behavior

Stage 44–45 tadpoles were placed in MBSH solution in a polystyrene Petri dish (100 × 15 mm) positioned directly over a 22 in computer monitor (ViewSonic 1080 Full HD, 10 M:1 contrast ratio), which lay horizontally within a darkened box. For each experiment, eight animals at a time were filmed using a digital camcorder (SONY HDR-CX250B), while the screen displayed dark moving dot stimuli (10 mm dia., 2 cm s$^{-1}$) following a ballistic random walker trajectory (*Molinàs-Mata et al., 1996*). A pre-test period of 3 min, with dots having the same luminance as the background (invisible dots) was used to establish the frequency of random chance false positive dot avoidance behaviors. During the test period, lasting 5 min, the screen displayed either a white background of high intensity (photopic, 190 ± 4.2 cd m$^{-2}$) or of low intensity (mesopic, 0.65 ± 0.05 cd m$^{-2}$), with dark dots of different luminances for each trial. Custom MATLAB 'Tadtracker' code (available for download at http://ruthazerlab.mcgill.ca/local.htm) was created to track each tadpole independently in the recorded videos and measure their reactions by correlating their locations, relative to the dots, with changes in their swimming speed to the dot coordinates. Tadpole positions were calculated based on image contrast to background with the software always identifying the eight objects fitting the criteria for a tadpole based on size and interpolation of previous and subsequent positions, and excluding the positions of known moving dot stimuli. Velocities were measured between each frame at 30 Hz video rate and then Gaussian smoothed to reduce the effects of noise on the tracking. A potential dot avoidance event was counted whenever a dot came within 5 mm of a tadpole. A

successful avoidance event was logged if swimming velocity changed by at least 24 mm s$^{-1}$ within 0.25 s of initial encounter. This threshold was determined empirically based on the acquisition frame-rate and baseline rates of tadpole movement as one which unambiguously constituted a reaction to the visual environment. We noted that histograms of tadpole swimming speeds showed a dual power law distribution ranging from 0 to 200 mm s$^{-1}$ with a critical point at 24 mm s$^{-1}$.

## Western blot for pNKCC1 and pSPAK

To analyze the changes in NKCC1 and SPAK phosphorylation, samples were prepared of stage 42–45 *Xenopus laevis* tadpole eyes. Eyes were homogenized in the extraction buffer (10 mM HEPES/NaOH pH 7.4, 150 mM NaCl, 2 mM EDTA pH 8.0, 1% NP40) with protease inhibitors (Calbiochem Protease inhibitor Set V, EDTA-free). Western blot analysis for NKCC1 phosphorylation was performed with Biorad wet transfer system and PVDF membrane, using the primary sheep polyclonal anti-human antibody against Thr phosphorylation sites 212, 217 of NKCC1 (MRC-PPU Reagents, antibody S063D). S063D was used at 0.5 µg/ml in 2% milk-TBS with 0.05% Tween, and before applying it to the blot, 30 min pre-incubation with 5 µg/ml of non-phosphorylated epitope peptide was performed, as per the recommendations of the manufacturer to minimize the prevalence of the antibodies in the polyclonal mix that react with the non-phosphorylated epitope. For total NKCC1 measurement, sheep anti-Dog fish polyclonal antibody (S841B, MRC PPU Reagents) was used, 0.2 µg/ml in 2.5% milk-PBS with 0.1% Triton. As a secondary antibody for S063D and S841B, Abcam rabbit anti-sheep IgG-HRP (RRID:AB_955453) was used, 1:30 000 in 2% milk TBS-T. For detection of SPAK phosphorylation, primary Millipore antibody 07–2273 p-SPAK Ser373/pOSR Ser325 (RRID:AB_11205577) was used, 1:7000 in 5% BSA, and as a secondary antibody, goat anti-rabbit HRP (RRID: AB_2307391), 1:30 000 in 5% milk TBS-tween. In all cases, blots were incubated in primary antibody overnight at 4C. Secondary antibodies were incubated at room temperature for 1 hr. The blots were developed with Immobilon Western Chemiluminescent HRP Substrate (WBKLS0500).

## Analysis

All statistical analyses were performed using Prism 6.0 (GraphPad, RRID:SCR_002798) and all data are reported and graphed as mean ± SEM. Data were tested for normality using a Shapiro-Wilk test and outliers, as identified by Grubb's test, were excluded from analysis. Correction for multiple comparisons following ANOVA were made using the Holm-Sidak post-hoc test with multiplicity corrected p-values reported. For one-sample t-tests, we tested whether fold-change was significantly different from one-fold (i.e. no change). Sample size in whole-cell recording experiments was based on values previously found sufficient to detect significant changes in retinotectal synaptic strength in past studies from the lab. For extracellular recordings, an N of 10 independent recordings from 10 animals was set in advance to provide sufficient statistical power while trying to minimize the number of animals sacrificed. Behavioral studies and RGC recording experiments were not performed blind, but analysis used automated software algorithms to present randomized stimuli and applied uniform criteria to all samples, limiting potential bias.

## Additional information

### Funding

| Funder | Grant reference number | Author |
| --- | --- | --- |
| Épilepsie Canada | Postdoctoral award | Loïs S Miraucourt |
| Natural Sciences and Engineering Research Council of Canada | CREATE Neuroengineering Training Program | Loïs S Miraucourt<br>Delphine Gobert<br>Jean-François Desjardins<br>Mari Sild<br>Perry Spratt<br>Paul W Wiseman<br>Edward S Ruthazer |
| Fonds de Recherche du Québec - Santé | Research chair, postdoctoral fellowship | Jennifer Tsui<br>Delphine Gobert<br>Edward S Ruthazer |

| Canadian Institutes of Health Research | Operating grants | Edward S Ruthazer |

The funders had no role in study design, data collection and interpretation, or the decision to submit the work for publication.

## Author contributions

LSM, Conceived and designed the project, Performed the experiments, Wrote the manuscript, Analysis and interpretation of data; JT, Conceived and designed the project, Performed the experiments, Analysis and interpretation of data, Drafting or revising the article; DG, MS, Performed the experiments, Conception and design, Analysis and interpretation of data, Drafting or revising the article; J-FD, Performed the experiments, Designed and created the tadpole tracking software and analyzed behavioural data, Contributed unpublished essential data or reagents; AS, Performed the experiments, Conception and design, Drafting or revising the article; PS, Performed the experiments, Analysis and interpretation of data, Drafting or revising the article; AC, YDK, Modified and tested the Clomeleon constructs and directed their use and the interpretation of data, Conception and design, Drafting or revising the article, Contributed unpublished essential data or reagents; NM-A, Provided transgenic animals, Helped design experiments in which they were used, Drafting or revising the article, Contributed unpublished essential data or reagents; PWW, Supervised the experiments, Conception and design, Analysis and interpretation of data, Drafting or revising the article, Contributed unpublished essential data or reagents; ESR, Conceived and designed the project, Supervised the experiments, Wrote the manuscript, Analysis and interpretation of data

## Author ORCIDs

Loïs S Miraucourt, http://orcid.org/0000-0003-4812-6342
Edward S Ruthazer, http://orcid.org/0000-0003-0452-3151

## Ethics

Animal experimentation: This study was performed in strict accordance with the recommendations in the Canadian Council on Animal Care. All animals were handled according to animal care committee protocols (#5071) approved by the Animal Care Committees of the Montreal Neurological Institute and McGill University.

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
