## [Decision Letter]

Thank you for submitting your article "Endocannabinoid signaling enhances visual responses through modulation of chloride levels in retinal ganglion cells" for consideration by *eLife*. Your article has been reviewed by two peer reviewers, and the evaluation has been overseen by Gary Westbrook as the Senior Editor. The following individuals involved in review of your submission have agreed to reveal their identity: Jeff Diamond (Reviewer #1); Rowland Taylor (Reviewer #2). The reviewers have discussed the reviews with one another and the Senior Editor has drafted this decision to help you prepare a revised submission..

Summary:

This is an interesting, thorough paper showing how cannabinoids enhance the sensitivity of visual signaling in the *Xenopus* visual system. The authors show that CB1 receptors are expressed in the *Xenopus* tadpole retina, including ganglion cells, and that CB1 receptor activation by exogenous or endogenous ligands enhances light-evoked spike responses in ganglion cells. Enhanced responses are not observed in bipolar or amacrine cells, indicating an effect on ganglion cells. The enhancement requires glycine receptor-mediated inhibition and is mimicked and occluded by blockade of the chloride transporter NKCC1. They go on to show that CB1 inhibits NKCC1 via AMP-dependent kinase. This reduction in NKCC1 activity lowers intracellular chloride, hyperpolarizing the ganglion cell membrane potential. The authors argue that this hyperpolarization actually enhances spiking response to light stimulation by reducing sodium channel inactivation. They show that CB1 activation regulates light-evoked firing in ganglion cells, enhances responses and broadens receptive fields in the tectum and enhances visually evoked behavioral responses under low light conditions. The experiments are cleanly performed and clearly presented. This is an excellent paper.

Essential revisions:

1)There were concerns regarding the evidence for reduced sodium channel inactivation (Figure 5). CB1 receptor activation causes a ~25% increase in light-evoked spike responses (Figure 1) and hyperpolarizes the resting membrane potential from -42 mV to -53 mV (perforated patch recordings, Figure 5). According to the authors' theory, these data would predict that a similar 11 mV hyperpolarization of the membrane potential would decrease sodium channel inactivation and increase (perhaps by 25%) spiking elicited by a depolarizing current step. The authors show that pre-hyperpolarizing the membrane leads to increased spikes in response to a test current step (Figure 5), but this experiment does not really test the idea as well as it could. First, the hyperpolarizing pre-steps cause a much greater change in membrane potential than that caused by CB1 activation (admittedly, these larger hyperpolarizations also cause larger changes in spiking). Also, there is a 25-ms interval between the end of the hyperpolarizing pre-pulse and the test step, and it's not clear how much change in inactivation occurs during this interval. It would seem more direct to hyperpolarize the membrane potential by 11 mV from rest with a long step, then immediately apply the test pulse to see if the spike response is enhanced 25% relative to the response with no pre-step. In addition, a voltage clamp experiment could be employed to show that, in these cells, sodium channel inactivation is substantially greater at -42 mV compared to -53 mV substantially reduces sodium channel inactivation. This second result is certainly expected, would not show causation, and may not be required; the first experiment (a modification of the protocol shown in Figure 5) is not necessarily critical but would strengthen the argument considerably.

2) Figure 2: The conclusion that the effects of CB1R agonists do not occur upstream of the RGCs seems a little strong, given the data. The data in Figure 2 presents the average effects of WIN across heterogeneous populations of RGCs, bipolar cells and amacrine cells. The only significant presynaptic WIN effect, a suppression of OFF bipolar activity (Figure 2), is expected to suppress OFF RGC responses, whereas an increase was observed (Figure 2). On could argue that such an apparent anomaly was due to network effects – reduced activity of some amacrine cells for example. A weakness in the approach is that there are many different types of bipolar (10-12), amacrine (40-50), and ganglion cells (20-30), and given the combinatorial possibilities, it could be difficult to drawn strong conclusions about the role of presynaptic circuit mechanisms from relatively small sample sizes. The conclusion from these experiments is that the effects of CB1R activation are unlikely to involved mechanisms presynaptic to the RGCs. This conclusion is critical, as it informs the interpretation of the subsequent analysis. Direct analysis of inputs to ganglion cells could help resolve this issue (see point 8).

3) Figure 3: The effects of the GABAergic and glycinergic antagonists seem unexpected (Figure 3). Generally, global suppression of inhibition might be expected to increase network excitability, yet neither GABAergic nor glycinergic block had much effect on light-evoked RGC spiking. The data suggest the possibility of polysynaptic network effects, e.g. serial interactions between GABAergic and glycinergic interneurons, which are known to occur in the retina. How might such effects impact the interpretation of the data? Have the authors applied GABAergic and glycinergic antagonists together?

4) Figure 3: The GABAergic antagonists had no effect on the WIN responses. If the CB1R effects are produced by changes in E_Cl within RGCs, then one might expect to see effects when suppressing either GABA or glycine receptors, as both receptors activate chloride channels in RGCs. This issue is addressed in the Discussion, and it is suggested that CB1Rs have direct, specific effects on glycine but not GABA receptors. No data is presented to support this suggestion. Doesn't this explanation also imply that the major effect of CB1R activation is via receptor modulation rather than changes in internal [Cl]?

5) Results section, subsection “CB1R-mediated enhancement of RGC fiing requires glycinergic transmission and NKCC1 activity”: "[…]and reversed (Figure 3) the enhancement caused by WIN 55,212-2 application." The change from +WIN to +WIN+Strychnine in Figure 3 does not appear to be significant. The text should be revised to more accurately reflect the data. If we are reading the table in the statistical form correctly, the +WIN condition in 3f is also not significant (P=0.088)?

6) The results for Figure 3 are difficult to follow. It would help to more clearly state the hypothesis at this point. If CB1R activation inhibits NKCC1, which reduces internal [Cl] in RGCs, then the CB1R agonist WIN should hyperpolarize RGCs and reduce spiking, which seems counter the data in Figure 2. The hypothesis to explain this apparent contradiction comes later, but it would help to introduce it earlier.

7) The lack of any significant effects for the ON-responses, shown in Figure 3—figure supplement 1 detracts from overall confidence in the findings. The currently available data do not appear to support applicability of the proposed model to ON-pathway signaling.

8).The central conclusion, that CB1Rs change RGC excitability by reducing internal [Cl], relies on the presence of a tonic inhibitory input from glycinergic amacrine cells to RGCs under physiological conditions. The conclusions would be stronger if the authors recorded directly from RGCs in an intact retinal preparation and showed the expected effects on membrane potential and excitability. Such experiments would also provide more direct evidence for the possible involvement, or not, of presynaptic mechanisms (see point 1).

---

## [Author Response]

*Essential revisions:*

*1)There were concerns regarding the evidence for reduced sodium channel inactivation (Figure 5). CB1 receptor activation causes a ~25% increase in light-evoked spike responses (Figure 1) and hyperpolarizes the resting membrane potential from -42 mV to -53 mV (perforated patch recordings, Figure 5). According to the authors' theory, these data would predict that a similar 11 mV hyperpolarization of the membrane potential would decrease sodium channel inactivation and increase (perhaps by 25%) spiking elicited by a depolarizing current step. The authors show that pre-hyperpolarizing the membrane leads to increased spikes in response to a test current step (Figure 5), but this experiment does not really test the idea as well as it could. First, the hyperpolarizing pre-steps cause a much greater change in membrane potential than that caused by CB1 activation (admittedly, these larger hyperpolarizations also cause larger changes in spiking). Also, there is a 25-ms interval between the end of the hyperpolarizing pre-pulse and the test step, and it's not clear how much change in inactivation occurs during this interval. It would seem more direct to hyperpolarize the membrane potential by 11 mV from rest with a long step, then immediately apply the test pulse to see if the spike response is enhanced 25% relative to the response with no pre-step. In addition, a voltage clamp experiment could be employed to show that, in these cells, sodium channel inactivation is substantially greater at -42 mV compared to -53 mV substantially reduces sodium channel inactivation. This second result is certainly expected, would not show causation, and may not be required; the first experiment (a modification of the protocol shown in Figure 5) is not necessarily critical but would strengthen the argument considerably.*

We had performed gramicidin perforated patch recordings of dissociated RGCs to provide electrophysiological validation of our in vivo Clomeleon imaging data. These recordings also had the added benefit of demonstrating that isolated RGCs without the intact retinal circuit to provide synaptic input show a cell-autonomous shift in V_m_ (old Figure 5, new Figure 6) as predicted. Our attempts to perform gramicidin perforated patch recordings in vivo were unsuccessful due to the well-known challenges associated with this technique.

Dissociated cultured neurons are unlikely to exhibit identical properties to intact neurons in situ. For example, we found these cultured cells to be considerably less prone to spiking than intact RGCs in the retina. In addition, we reported V_m_ without correcting for the liquid junction potential. Overall we believe the precise membrane voltages measured in these in vitro recordings can be used to interpret relative changes but not absolute voltages that likely occur in the intact retina in vivo in response to CB1R activation.

In Figure 6 (old Figure 5) we showed that a hyperpolarizing prepulse increases the firing rates of RGCs in response to depolarizing current injected 25 ms later. This protocol was adapted from a *Neuron* article by Weick & Demb (2011). The reviewers indicated that, while not necessarily critical, our case might have been made more directly by simply reducing membrane voltage by the “predicted” 11mV during the prepulse and omitting the 25 ms interval. We have therefore implemented a version of this experiment in the accompanying figure, which demonstrates that the outcome is not substantially different when the 25 ms interval is removed (n =6). Indeed the example shown on the right illustrates that a 10 mV hyperpolarizing prepulse can increase spike rates by about 20%. We do not know precisely where on this curve the cells actually would sit prior to patching, but the data are entirely consistent with spike rate changes of the magnitude we report throughout the paper.

Author response image 1.**DOI:**
http://dx.doi.org/10.7554/eLife.15932.017

*2) Figure 2: The conclusion that the effects of CB1R agonists do not occur upstream of the RGCs seems a little strong, given the data. The data in Figure 2 presents the average effects of WIN across heterogeneous populations of RGCs, bipolar cells and amacrine cells. The only significant presynaptic WIN effect, a suppression of OFF bipolar activity (Figure 2), is expected to suppress OFF RGC responses, whereas an increase was observed (Figure 2). On could argue that such an apparent anomaly was due to network effects – reduced activity of some amacrine cells for example. A weakness in the approach is that there are many different types of bipolar (10-12), amacrine (40-50), and ganglion cells (20-30), and given the combinatorial possibilities, it could be difficult to drawn strong conclusions about the role of presynaptic circuit mechanisms from relatively small sample sizes. The conclusion from these experiments is that the effects of CB1R activation are unlikely to involved mechanisms presynaptic to the RGCs. This conclusion is critical, as it informs the interpretation of the subsequent analysis. Direct analysis of inputs to ganglion cells could help resolve this issue (see point 8).*

We agree with the reviewers here but see no contradiction with our thesis. Our argument is not that there are absolutely no upstream changes that could contribute to RGC firing –as the reviewers point out the complexity of the retina makes it virtually impossible to exclude such a contribution. Indeed we discuss such upstream mechanisms (e.g., cannabinoid modulation of glyRs) at length in the Discussion section. From the beginning our objective has simply been to provide multiple lines of evidence for a novel form of cannabinoid-mediated regulation of excitability and to make a compelling case that the eCB system in the retina indeed exploits this mechanism to drive network and behavioral changes.

Our data show that this novel mechanism based on CB1R-dependent chloride regulation in RGCs is both necessary and sufficient to drive the changes in RGC responsiveness, but they cannot exclude all possible contributions by other factors. Nonetheless, in further support of the idea that direct action of cannabinoids on RGCs is the primary influence, we have now added the experiment proposed below in point 8 (new Figure 3). It shows that under the recording conditions used, CB1R-dependent effects on visually evoked RGC inputs are unlikely to fully account for the spike enhancement observed.

*3) Figure 3: The effects of the GABAergic and glycinergic antagonists seem unexpected (Figure 3). Generally, global suppression of inhibition might be expected to increase network excitability, yet neither GABAergic nor glycinergic block had much effect on light-evoked RGC spiking. The data suggest the possibility of polysynaptic network effects, e.g. serial interactions between GABAergic and glycinergic interneurons, which are known to occur in the retina. How might such effects impact the interpretation of the data? Have the authors applied GABAergic and glycinergic antagonists together?*

We did not perform an experiment in which gabazine and strychine were applied together. However, in experiments that were not included in the paper, we did find that picrotoxin, which dually targets both GABA-ARs and extrasynaptic GlyRs, partially blocks the CB1R-mediated change in firing rates. We chose not to include the picrotoxin data in this manuscript because the effect was less clear than strychnine alone, and moreover the action of picrotoxin on extrasynaptic GlyRs is not well-appreciated by the general public. As a result, readers tended to be more confused by this experiment. A figure is included here for your evaluation.

Author response image 2.**DOI:**
http://dx.doi.org/10.7554/eLife.15932.018

*4) Figure 3: The GABAergic antagonists had no effect on the WIN responses. If the CB1R effects are produced by changes in E_Cl within RGCs, then one might expect to see effects when suppressing either GABA or glycine receptors, as both receptors activate chloride channels in RGCs. This issue is addressed in the Discussion, and it is suggested that CB1Rs have direct, specific effects on glycine but not GABA receptors. No data is presented to support this suggestion. Doesn't this explanation also imply that the major effect of CB1R activation is via receptor modulation rather than changes in internal [Cl]?*

We agree that there is probably some contribution of direct GlyR modulation, as we discuss in our Discussion section. However, all our CB1R-mediated effects could be mimicked and occluded by bumetanide application, suggesting that chloride regulation in RGCs specifically is both necessary and sufficient to underlie the CB1R-mediated enhancement of RGC firing rates.We interpret the observation that strychnine, but not gabazine, prevents this effect to indicate that there is substantially more tonic activation GlyRs than GABA-A receptors at rest

*5) Results section, subsection “CB1R-mediated enhancement of RGC fiing requires glycinergic transmission and NKCC1 activity”: "[…]and reversed (Figure 3) the enhancement caused by WIN 55,212-2 application." The change from +WIN to +WIN+Strychnine in Figure 3 does not appear to be significant. The text should be revised to more accurately reflect the data. If we are reading the table in the statistical form correctly, the +WIN condition in 3F is also not significant (P=0.088)?*

Here in Figure 4 (old Figure 3) we used one-tailed tests as we had already established in Figure 2 that the relevant effect of interest is an increase in firing rates. Thus, we only sought to test if treatment permitted or prevented the increased firing – the p-values in the table should therefore be divided by 2 to obtain one-tailed values. In fact both the change upon addition of WIN and the decrease upon subsequent addition of strychnine are significant by this criterion.

This was not made sufficiently clear in the original text, which has now been revised to specify that a one-tailed test was applied.

*6) The results for Figure 3 are difficult to follow. It would help to more clearly state the hypothesis at this point. If CB1R activation inhibits NKCC1, which reduces internal [Cl] in RGCs, then the CB1R agonist WIN should hyperpolarize RGCs and reduce spiking, which seems counter the data in Figure 2. The hypothesis to explain this apparent contradiction comes later, but it would help to introduce it earlier.*

Thank you for this suggestion. We have added a paragraph to introduce this concept prior to presenting the data in Figure 4 (old Figure 3).

*7) The lack of any significant effects for the ON-responses, shown in Figure 3—figure supplement 1 detracts from overall confidence in the findings. The currently available data do not appear to support applicability of the proposed model to ON-pathway signaling.*

The effect is more consistent for the OFF pathway, but not absent from the ON responses (e.g., Figure 2). This probably reflects the fact that in *Xenopus* light-OFF responses are generally more robust than responses to light-ON stimuli.

*8) The central conclusion, that CB1Rs change RGC excitability by reducing internal [Cl], relies on the presence of a tonic inhibitory input from glycinergic amacrine cells to RGCs under physiological conditions. The conclusions would be stronger if the authors recorded directly from RGCs in an intact retinal preparation and showed the expected effects on membrane potential and excitability. Such experiments would also provide more direct evidence for the possible involvement, or not, of presynaptic mechanisms (see point 1).*

Not surprisingly, we were unable to perform reliable in vivo gramicidin perforated patch recordings, which are notoriously difficult. However we have added two new sets of experiments involving in situ whole-cell recordings from RGCs:

1) To provide further evidence for a CB1R-mediated inhibition of NKCC1, leading to reduced intracellular Cl^-^, we performed in vivo whole cell recordings from RGCs while puffing glycine in order to test for a shift in the glyR I-V curve (Figure 6). Because any changes in intracellular Cl^-^ must overcome the Cl^-^ loading by the whole-cell recording pipette, this method does not directly demonstrate the changes in intracellular Cl^-^ levels that would occur in intact cells, but rather reveals the activation of Cl^-^ transporters that underlie these changes. We demonstrate that CB1R activation produces a leftward shift in the I-V curve that mimics that produced by directly inhibiting NKCC1 with bumetanide. This provides additional evidence that CB1R activation inhibits NKCC1 in RGCs.

2) We have also more directly examined the effects of cannabinoid application on synaptic inputs onto RGCs by performing in vivo whole-cell recordings during visual stimulation (Figure 3). These experiments reveal that retinal CB1R activation decreases visually evoked excitatory synaptic drive onto RGCs, consistent with the GCaMP6 imaging data in Figure 3 (old Figure 2). An IPSC reduction is also seen. Thus in this case, CB1R agonists appear to carry out their classic role of decreasing synaptic release probability of bipolar and amacrine cells. This is unlikely to account for the increased firing rates of RGCs that we observe, especially in light of the fact that we know that the driving force for Cl^-^ in intact neurons is enhanced, which would counteract the decrease in evoked synaptic IPSCs.